# WhodunitBench: Evaluating Large Multimodal Agents via Murder Mystery Games

**Junlin Xie**[1,2,♡], **Ruifei Zhang**[1,2,♡], **Zhihong Chen**[1,2,♣], **Xiang Wan**[2], **Guanbin Li**[3,4,5,♣]

[1] The Chinese University of Hong Kong, Shenzhen
[2] Shenzhen Research Institute of Big Data
[3] Sun Yat-sen University [4] Peng Cheng Laboratory
[5] GuangDong Province Key Laboratory of Information Security Technology
{junlinxie,ruifeizhang,zhihongchen}@link.cuhk.edu.cn
wanxiang@sribd.cn, liguanbin@mail.sysu.edu.cn

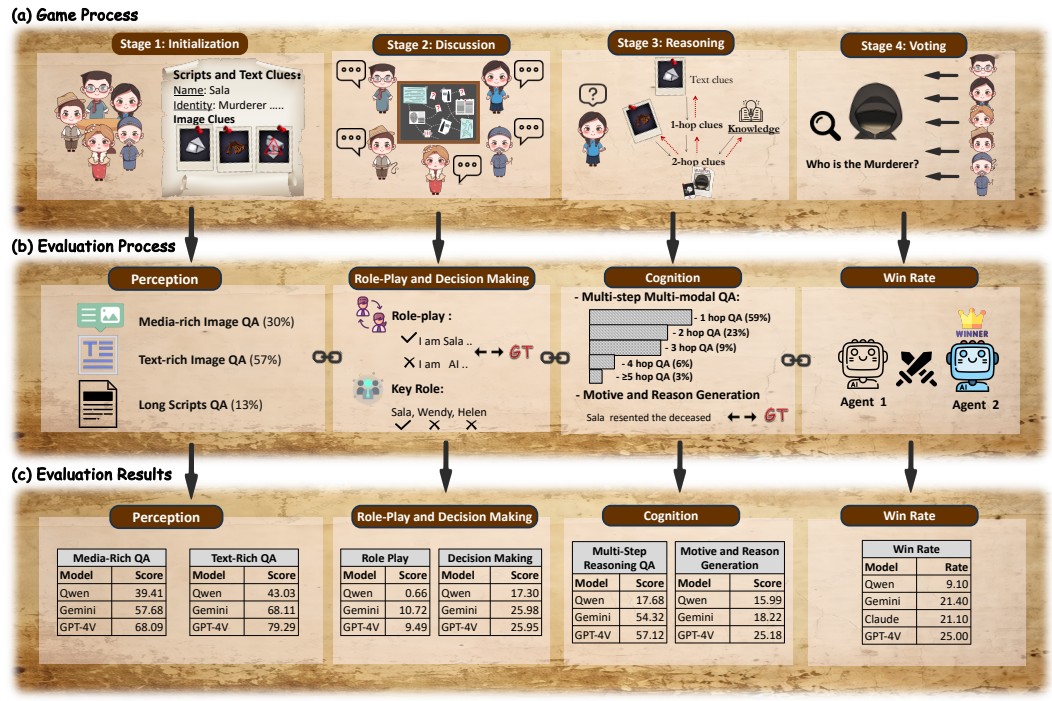

Figure 1: Overview of our proposed WhodunitBench, a benchmark for large multi-modal agents simulated from *murder mystery games*: (a) the introduction of the game process; (b) the evaluable capabilities and corresponding assessment methods derived from the game; (c) the evaluation results.

## Abstract

Recently, large language models (LLMs) have achieved superior performance, empowering the development of large multimodal agents (LMAs). An LMA expected to perform practical tasks must possess a range of capabilities, including multimodal perception, interaction, reasoning, and decision-making skills. However, existing benchmarks are limited in assessing compositional skills and actions

---

♡ Equal contribution ♣ Corresponding authors.

38th Conference on Neural Information Processing Systems (NeurIPS 2024) Track on Datasets and Benchmarks.

demanded by practical scenarios, where they primarily focused on single tasks and static scenarios. To bridge this gap, we introduce WhodunitBench, a benchmark rooted from *murder mystery games*, where players are required to utilize the aforementioned skills to achieve their objective (i.e., identifying the 'murderer' or hiding themselves), providing a simulated dynamic environment for evaluating LMAs. Specifically, WhodunitBench includes two evaluation modes. The first mode, the arena-style evaluation, is constructed from 50 meticulously curated scripts featuring clear reasoning clues and distinct murderers; The second mode, the chain of evaluation, consists of over 3000 curated multiple-choice questions and open-ended questions, aiming to assess every facet of the murder mystery games for LMAs. Experiments show that although current LMAs show promising performance in basic perceptual tasks, they are insufficiently equipped for complex multi-agent collaboration and multi-step reasoning tasks. Furthermore, the full application of the theory of mind to complete games in a manner akin to human behavior remains a significant challenge. We hope this work can illuminate the path forward, providing a solid foundation for the future development of LMAs. Our WhodunitBench is open-source and accessible at: `https://github.com/jun0wanan/WhodunitBench-Murder_Mystery_Games`.

# 1   Introduction

Large multimodal agents (LMAs) [29, 21, 27, 11] are systems capable of perceiving its environment and making decisions based on these perceptions to achieve specific goals within the multimodal context driven by large language models (LLMs). LMAs are anticipated to handle *diverse* and *challenging* tasks that demand a broad range of capabilities, including low-level multimodal perception, high-level cognition (e.g., multi-step reasoning), role-playing for interactive engagement and deliberative decision-making.

Given these diverse capabilities, the evaluation of LMAs varies widely across research domains. Some studies prioritize the agents' competency in executing complex internet-based tasks [8, 10, 16, 30], while others focus on assessing their reasoning and decision-making abilities [14, 15, 26, 6]. Additionally, a significant body of research explores these agents' capacities for long-term planning and execution [35, 28, 33]. However, it is challenging to design such an evaluation benchmark to evaluate the various capabilities of LMAs within the same environment. We categorize the capabilities into the following four classes:

- **Multi-modal Perception** is the most basic ability for LMAs, which requires LMAs to perceive information from the multimodal environment (e.g., vision and language).
- **Interaction** requires LMAs, whether through role-playing or direct engagement, to communicate with the environment or other agents to gather essential information for task completion.
- **Reasoning** requires LMAs to combine their internal knowledge with newly gathered information to perform long-chain, multi-step reasoning.
- **Decision Making and Goal Achievement** requires LMAs to establish clear goals and make independent decisions in response to environmental changes. This autonomous decision-making is crucial for effectively navigating and completing tasks in dynamic settings.

Interestingly, murder mystery games [5, 12, 3], a genre of party games, offer a unique and covert opportunity to evaluate LMAs across the forementioned dimensions.[*] As illustrated in Figure 1, a murder mystery game unfolds in a virtual world crafted by multiple players, with the goal being to identify the 'murderer' through the following procedure:

- **Initialization Phase**: Players need to perceive multimodal information , including extensive script text and various types of image clues, while role-playing their assigned characters to present this information.
- **Discussion Phase**: Players need to role-play their own roles to interact with the environments or other players to get more clues. During this process, they need to perform decision making to assess the authenticity of information and to gather more details to supplement and refine the tasks required for each role.

---

[*]`https://en.wikipedia.org/wiki/Murder_mystery_game`

Table 1: Detailed comparative analysis of our benchmark with others across multiple dimensions. Specifically, "Incomplete Information" refers to cases where an agent's information includes only a portion of what is required for reasoning, with the remaining information needing to be acquired through effective interactions. Meanwhile, "Online Competition" denotes direct, real-time, head-to-head matches between agents in a dynamic environment.

| Benchmarks | Multi-Modal | Multi-step Reasoning | Role-play | Reasoning Type | Evaluation Type |
|---|---|---|---|---|---|
| DDD [26] | ✗ | ✗ | ✓ | Incomplete Information | Isolated Evaluation |
| AVALONBENCH [15] | ✗ | ✗ | ✓ | Incomplete Information | Online Competition |
| GAIA [16] | ✓ | ✓ | ✗ | ✗ | Isolated Evaluation |
| VisualWebArena [13] | ✓ | ✗ | ✗ | ✗ | Isolated Evaluation |
| WorldQA [34] | ✓ | ✓ | ✗ | Complete Information | Isolated Evaluation |
| MCOT [7] | ✓ | ✓ | ✗ | Complete Information | Isolated Evaluation |
| Rolellm [23] | ✓ | ✗ | ✓ | ✗ | Isolated Evaluation |
| SOTOPIA [36] | ✗ | ✗ | ✓ | ✗ | Isolated Evaluation |
| WhodunitBench (ours) | ✓ | ✓ | ✓ | Incomplete Information | Online Competition & Chain of Evaluation |

- **Reasoning Phase**: Players need to reason over the information collated from the previous two phrases, always involving complex multi-step multi-modal reasoning.
- **Voting Phase**: Ultimately, through a voting process involving all players, the 'murderer' is determined. This can evaluate if the players achieve their goals (i.e., identifying the 'murderer' or hiding themselves).

Therefore, based on this, we introduce a comprehensive benchmark via murder mystery games designed to evaluate LMAs (named WhodunitBench) in this paper. Table 1 presents the primary characteristics of our benchmark in comparison to others. Specifically, we propose two evaluation modes: (1) **Arena-style Evaluation**, which simulates real gameplay by having agents act as players in one-on-one online competitions, uses their win rate as the primary evaluative metric. (2) **Chain of Evaluation**, which provides a comprehensive analysis of agent performance by designing and annotating over 3,000 multiple-choice questions and brief answer sentences. In this evaluation, each metric is designed to align with the game environment while comprehensively supplementing previous assessments. We select five representative LMAs, including Yi-Vision [32], Qwen-VL-Plus [4], Gemini-pro-vision [19], Claude-Opus [2] and GPT-4V [1] and conduct extensive experiments on our WhodunitBench . Experimental findings reveal that even the advanced GPT-4V [1], which attains the highest win rate in the online arena, still encounters challenges in successfully completing this game. **Hallucinations [17],** failure to truly understand the script, and difficulty in immersing into roles are its primary error manifestations. Our Chain of Evaluation (CoE) offers more insights for researchers, highlighting that while LMAs typically perform well in basic perception, they struggle with complex multi-modal reasoning and effective interaction within role-playing scenarios. Ultimately, the contributions of our paper are three-fold:

- We propose to use *murder mystery games* as the environments to assess a variety of abilities of LMAs. To this end, we design a benchmark, called WhodunitBench, consisting of two modes: an online battle arena and a chain of evaluation.
- We curate evaluation samples in two modes: 50 scripted scenarios using win rate for direct confrontation between LMAs, and over 3,000 multiple-choice and open-ended questions to quantify specific capabilities, complementing the win rate assessment with detailed skill evaluations.
- Experiments conducted on WhodunitBench demonstrate that existing state-of-the-art LMAs struggle in dynamic scenarios and complex composition tasks. Against our naively designed agent, these agents achieve a maximum win rate of only 25%, and their scores for role-play interaction barely exceed 20 points.

## 2 Related Work

**Evaluating Agent.** As LLMs become increasingly prevalent, the development of intelligent agents and the benchmarks for evaluating them continue to evolve [16, 8, 10, 15]. Previous benchmarks have primarily focused on simple yet tedious web-based tasks [16, 30, 8] aimed at developing agents

capable of managing repetitive aspects of human online activities. Besides, environments such as "Werewolf" [15, 22] are used to assess agents' strategic and decision-making skills, while other benchmarks [28, 35] evaluate long-term strategy and adaptability in specialized scenarios. In contrast, our proposed benchmark evaluates agents in realistic scenarios, where they must simultaneously employ multiple skills rather than focusing on a single ability in a controlled lab environment. More importantly, these skills closely mirror those humans rely on when completing tasks in the real world. This includes the perception and understanding of multimodal content, gathering additional information through interactions with the surrounding environment or other individuals, and finally, integrating this information with prior knowledge to carry out multi-step analysis, reasoning, and decision-making under incomplete information to accomplish their tasks.

**Evaluating LMAs on Gaming Platforms.** Games [11, 9], with their simple rules, clear standards, controllable difficulty, and limited scope for action or observation, are increasingly being used as benchmarks for evaluation agents. In addition to the Werewolf-style text games previously mentioned, studies have also explored using games like "Red Dead Redemption II" [18] and various open-world environments [31, 24] to evaluate the capabilities of LMAs. Employing these games for testing generally demands significant resources and time. Some researchers also have suggested employing murder mystery games as a more efficient alternative for testing [26]. They primarily assessed text-based agents using relatively straightforward evaluation methods, focusing on multiple-choice questions. In contrast, our evaluation system not only offers two distinct assessment methods but also integrates a range of question types in the second method, particularly emphasizing multi-step multi-modal long-chain reasoning questions. This comprehensive evaluation system fully leverages the scripted murder mystery platform to test agents' abilities in dynamic, information-incomplete environments, closely mirroring human performance.

# 3 WhodunitBench: Construction

In this section, we describe the construction of WhodunitBench, which features an online competitive arena that simulates a realistic gameplay experience, as well as the CoE framework designed to assess LMAs' capabilities through a sequence of "Perception - Role-playing Interaction - Cognition" aligned with the respective stages of gameplay.

## 3.1 Constructing Arena

**Data Collection:** The construction of different games relies on diverse scripts, making the selection and collection of these scripts particularly crucial. We enlisted the expertise of seasoned murder mystery game experts to ensure the quality and applicability of the selected scripts. These scripts were sourced primarily from industry-recognized creative teams and platforms. We established clear selection criteria focused on three key aspects:

- **Scientific Integrity**: We have systematically excluded scripts incorporating metaphysical elements, particularly temporal displacement and consciousness transference. This methodological approach ensures that murder mysteries within these scripts remain grounded in empirical logic and scientific principles, thus maximizing operational viability and narrative credibility.
- **Content Complexity**: We chose scripts with a higher degree of reasoning complexity to thoroughly test the deductive capabilities of LMAs.
- **Logical Coherence**: We ensured all scripts were logically sound, with evidence and clues distributed in a balanced and reasonable manner.

**Data Quality Control:** We conducted a systematic review and optimization of the 50 real scripts collected. Initially, we ensured that the extracted script sections were complete, and we verified the fluidity and grammatical correctness of the text. Subsequently, we confirmed the completeness and integrity of the visual and textual clues within the scripts. Lastly, we examined the consistency of the timeline and the sequence of events, ensuring the logical coherence and rational progression of the plot. Following the comprehensive selection and rigorous review processes delineated above, we curated and refined a total of 50 scripts that conformed to our stringent criteria. The distribution overview of the number of roles in the script is shown in Figure 2 (b). These scripts were utilized to construct the online competitive arena.

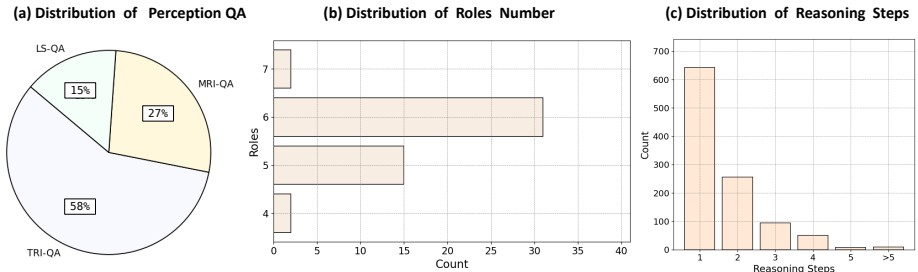

Figure 2: Statistics of the proposed dataset: (a) Distribution of perception QA; (b) Distribution of the number of roles in the scripts; (c) Distribution of reasoning steps for cognition assessments.

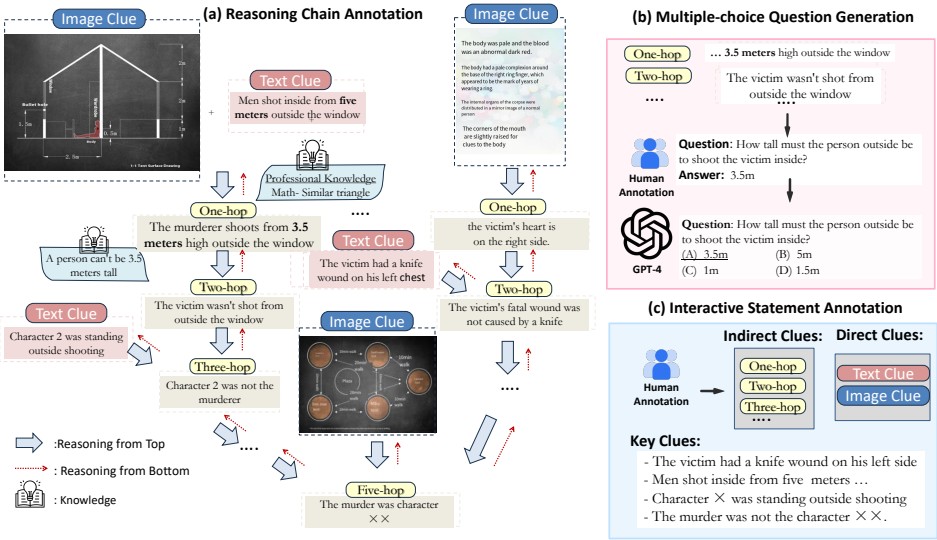

Figure 3: Data generation and annotation: (a) Examples of annotated ground truth reasoning chains; (b) Multiple-choice question generation process; (c) Interactive statement annotation process.

## 3.2 Constructing Chain of Evaluation (CoE) Dataset

### 3.2.1 Perception

**Question Types**: The evaluation of perception consists of **multiple-choice questions** categorized into three types: (1) Text-rich Image Questions (TRI-QA): These questions primarily involve text-based clues presented as images within the game, particularly those containing only text. (2) Media-rich Image Questions (MRI-QA): These questions primarily concern image clues within the game that contain both rich textual and visual elements. (3) Long Script Questions (LS-QA): These questions pertain to the textual content embedded within the game's script and role's script.

**Question Statistics**: There are a total of 1,911 multi-choice questions for perception assessment, categorized into 283 long script questions, 1,103 text-rich image questions, and 525 media-rich image questions. The distribution is illustrated in Figure 2(a).

### 3.2.2 Role-play Interaction

**Question Types**: To evaluate the role-play interaction capability of LMAs, we primarily annotated two types of data to serve as the ground truth for our assessment: (1) The first category comprises a collection of statements containing key clues in each script. (2) The second category consists of the core roles within each script.

**Question Annotation**: As shown in Figure 3(c), we compile all key clues necessary to solve each murder mystery in the script, encompassing both direct and indirect clues. These clues serve as ground truth statements for evaluating the effectiveness of LMAs in their role-play and for advancing the game's progression during the discussion phase. Additionally, we identify the core roles in each script, whose personal scripts contain critical clues for identifying the murderer. For instance, the clue "Character 2 was standing outside shooting" appears solely in Character 1's script, marking Character 1 as a core role.

### 3.2.3 Cognition

**Question Types**: The design of cognitive evaluation questions primarily consists of two types: (1) Multi-choice questions for multi-step reasoning assessments. (2) Open-ended questions to evaluate the accuracy of LMAs' analysis of murderer's motives and methods.

**Question Annotation**: Each script is accompanied by a truth manual that contains all clues essential for task completion. Annotators reference this manual, refining details to establish the final ground truth statements concerning the murderer's motive and methodology. The annotation process of multi-choice questions is divided into two stages, as shown in Figure 3(a) and (b): ① Construction of Reasoning Chains: we construct each reasoning step required to unravel the murder mystery within the script, leveraging the information provided in the truth manual and supplementing it with essential details. For example, to deduce that "The victim's fatal wound was not caused by a knife", we first identify key clues given directly in the game, such as images showing the victim's internal organs mirrored compared to a normal person. This information leads us to a 1-hop indirect but crucial clue: the victim's heart is on the right side. Further combining this with expert knowledge and textual clues about a knife wound on his left chest, we infer a 2-hop indirect clue: the knife wound was not fatal. Using this approach, we continuously pinpoint direct clues and deduce indirect ones, eventually linking them into a complete reasoning chain. ② Constructing multiple-choice questions: after building the reasoning chains, annotators use the content of these chain nodes to formulate tiered reasoning questions with correct answers. GPT-4 then creates distractor options based on these correct answers, matching their length to enhance confusion.

**Question Statistics**: We annotated 1,308 reasoning multiple-choice questions. The distribution across different levels is illustrated in Figure 2(c).

### 3.2.4 Data Quality Control

To improve our dataset's quality, we engaged three experts to perform data reviews based on specific standards. Any questions not meeting these standards will be refined. The standards are as follows: (1) If there are substantial informational gaps between nodes in the reasoning chain, intermediate steps will be added to maintain logical consistency; (2) For incorrect options generated by GPT-4, if they are too simplistic or clearly implausible within the problem's context, experts will manually revise them.

## 4 WhodunitBench: Arena-style Evaluation

WhodunitBench provides an online arena where LMAs compete in pairwise, faction-based matches, with win rates serving as the primary measure of success. Additionally, we record each multimodal agent's performance data in the arena, including their dialogue outputs and chosen actions.

### 4.1 Settings

**Agent Settings** We design two settings: one where a naive agent is defined as the murderer and each LMA competes against this naive agent, and another where selected LMAs engage in pairwise competitions. In the game, a multimodal agent in the non-murderer suspect faction, which comprises various roles, will adopt different roles to compete against the agent representing the murderer in the murderer faction. For **LMAs**, we selected five multimodal agents for evaluation: Yi-Vision [32], Qwen-VL-Plus [4], Gemini-pro-vision [19], Claude-Opus [2], GPT-4V [1]. For **Naive Agent**, we defined a naive agent that retrieves information about itself and provides output based on the search. When questioned by others, it finds and responds with content related to its role; if it finds nothing,

Table 2: Comparative benchmarking of LMAs in an online battle arena. Each cell (Row, Col) indicates the win rate of the Row agent against the Col agent. Note that we excluded instances where no clear winner was determined, including cases with API errors or draws in the voting for the murderer.

| Agent vs. Agent | Naive Agent | Yi-Vision | Qwen-VL-Plus | Gemini | Claude | GPT-4V | Avg. Win Ratio (%) |
|---|---|---|---|---|---|---|---|
| Yi-Vision | 12.0% | - | 10.0% | 14.7% | 11.6% | 7.1% | 11.1% |
| Qwen-VL-Plus | 9.1% | 6.8% | - | 16.2% | 11.4% | 9.5% | 10.6% |
| Gemini | 21.4% | 13.2% | 15.8% | - | 22.9% | 11.4% | 16.9% |
| Claude | 21.1% | 11.1% | 15.9% | 25.0% | - | 22.7% | 19.2% |
| GPT-4V | 25.0% | 18.2% | 23.3% | 29.5% | 25% | - | **24.2%** |
| **Avg. Loss Ratio (%)** | 17.7% | 12.3% | 16.3% | 21.4% | 17.7% | 12.7% | - |

it simply answers, "I don't know." Moreover, if in the murderer faction it is suspected of being the murderer and it retrieves information confirming this, it will immediately reveal its identity.

**Metric** In the arena, we use win rate and loss rate as the sole evaluation criteria. The non-murderer suspect faction wins by correctly identifying the murderer, whereas the murderer faction wins by evading identification. In the table 2, the rows represent the non-murderer suspect faction, and the columns represent the murderer suspect faction. We utilize the average win rate ($\frac{\text{win match}}{\text{total match}}$) and average loss rate($\frac{\text{loss match}}{\text{total match}}$) to assess LMAs' performance. If the average win rate is high or the average loss rate is low, it indicates that the agent is strong.

## 4.2 Results

We report the results in Table 2. We have the following observations: (1) **The overall win rate remains low.** Regardless of the type of multimodal intelligent agent assuming the role of the "Non-Murder," their win rates hover between 10% and 20%. This underscores the substantial challenges faced by all current advanced LMAs, including the latest iteration, GPT-4V, in achieving the objectives set out in the game. This suggests a significant gap in the performance capabilities of these agents when tasked with complex, goal-oriented tasks in dynamic environments; (2) **Stronger models do not necessarily perform better when playing the role of the murderer**. For instance, the Gemini model, regardless of its opponent, is most likely to be identified as the murderer. This may be because more capable models, realizing their role as the murderer, tend to over-communicate in an attempt to obscure the truth, which ironically makes them more susceptible to detection by other players. Conversely, less capable models, such as Qwen, might speak less frequently, making them less likely to be convicted in the game.

## 5 WhodunitBench: Chain of Evaluation (CoE)

In this section, we introduce the CoE Evaluation System, specifically designed to assess three core capabilities through a detailed framework of eight evaluation metrics, grounded in the annotated data from the previous section. With these design standards, we can not only systematically analyze and evaluate each agent's performance at various stages of the game but, more importantly, also provide a strong supplement to previous online competition assessments, showcasing each LMA's capabilities in greater detail. Table 3 presents the evaluation metrics for each LMA when playing a non-murder role against the naive agent we designed (acting as the murderer). Since the naive agent lacks certain "intelligence," the murderer does not interfere, allowing for a clearer demonstration of each LMA's performance across various capabilities.

### 5.1 Assessment Details

**Perceptual Ability Assessment**: To successfully complete the task, the agent must be able to perceive and comprehend a substantial amount of visual and textual information across various stages of the game, particularly during the initialization phase (as illustrated in the figure 1). These information

Table 3: Evaluating LMAs in non-murderer factions versus naive agents using the COE dataset revealed distinct outcomes.

| Model and Reasoning | Perception | | | Role-play | | Decision-Making | Cognition | | Avg |
| --- | --- | --- | --- | --- | --- | --- | --- | --- | --- |
| | LSU | TIU | MIU | RP | SPC | ITD | MMR | CMD | |
| Random | 25.00 | 25.00 | 25.00 | - | - | - | 25.00 | - | - |
| *Yi-Vision [32]* | | | | | | | | | |
| Direct[20] | 42.40 | 28.66 | 34.99 | 7.16 | 2.37 | 20.61 | 20.31 | 16.03 | 21.57 |
| COT[25] | 32.80 | 15.36 | 27.40 | 7.20 | 2.79 | 15.26 | 25.41 | 22.47 | 18.58 |
| *Qwen-VL-Plus [4]* | | | | | | | | | |
| Direct[20] | 38.40 | 43.03 | 39.41 | 7.15 | 0.66 | 17.30 | 17.68 | 15.99 | 22.45 |
| COT[25] | 36.00 | 51.36 | 46.50 | 7.09 | 0.76 | 20.61 | 22.03 | 13.59 | 24.74 |
| *Gemini [19]* | | | | | | | | | |
| Direct[20] | 92.00 | 68.11 | 57.68 | 7.45 | 10.72 | 25.98 | 54.32 | 18.22 | 41.81 |
| COT[25] | 88.80 | 57.78 | 57.84 | 7.22 | 10.79 | 19.08 | 57.39 | 19.20 | 39.76 |
| *Claude [2]* | | | | | | | | | |
| Direct[20] | 90.00 | 67.39 | 52.98 | 8.00 | 9.51 | 19.63 | 55.08 | 18.96 | 40.19 |
| COT[25] | 88.80 | 35.31 | 55.02 | 7.89 | 12.08 | 25.45 | 57.78 | 22.07 | 38.05 |
| *GPT-4V [1]* | | | | | | | | | |
| Direct[20] | 93.60 | 79.29 | 68.09 | 7.98 | 9.49 | 25.95 | 57.12 | 25.18 | 45.84 |
| COT[25] | 92.40 | 51.88 | 69.25 | 6.43 | 19.63 | 16.28 | 58.75 | 26.43 | 42.63 |

are typically referred to as mystery scripts and clues within the game. We have developed three categories of metrics for evaluation: (1) Text-rich image understanding (TIU): This metric assesses agents' proficiency in precisely interpreting and extracting clues from text-rich images, emphasizing their Optical Character Recognition (OCR) capabilities. It primarily utilizes the TRI-QA annotations from Section 3.2.1. (2) Media-rich image understanding (MIU): This metric evaluates how effectively agents integrate textual and visual elements to interpret and understand more complex clues within images, which may include diagrams, maps or residential layouts. It aims to gauge the agents' ability to navigate intricate visual cues that require both recognition and contextual comprehension. And it primarily utilizes the MRI-QA annotations from Section 3.2.1. (3) Long-script understanding (LSU): This metric evaluates agents' ability to process and extract critical information from lengthy texts, specifically the script content within the game, which sometimes exceeds tens of thousands of words in length. It primarily utilizes the LS-QA annotations from Section 3.2.1. Their scoring formula is defined as: $\text{Score}_{(\text{LSU, MIU, TIU})} = \frac{\text{Correct Questions per Category}}{\text{Total Questions per Category}}$.

**Strategic Decision-Making and Role-playing Assessment**: To evaluate the role-playing and interactive communication abilities of LMAs, we recorded their dialogues and performances from the online competition and assessed them using two metrics: (1) RP (Role-Playing) Index: This metric assesses the naturalness of the recorded agent dialogues with other roles. It is scored on a ten-point scale, with several criteria designed for GPT-4 to use in scoring. (2) SPC (Scenario Progression Capability) Index: This metric evaluates whether the agent's dialogue contributes to task completion (e.g., identifying the murderer's motive), rather than discussing irrelevant or off-topic content. The score is calculated using annotated ground truth statements from Section 3.2.2: $\text{Score} = \left( \frac{\text{Number of correct statements}}{\text{Total number of statements in the script}} \right) \times 100$. Additionally, we assess the decision-making ability of the agent during the discussion phase. Points will be awarded if the agent chooses to question previously identified key roles from Section 3.2.2 and deems this decision valuable. The calculation method is as follows: $\text{Score} = \left( \frac{\text{Number of key characters successfully questioned}}{\text{Total number of key characters in the game}} \right) \times 100$.

**Comprehensive Cognition Assessment**: To accurately identify the murder in the murder mystery, the agent must integrate various clues to perform complex reasoning across different levels. This evaluation primarily focuses on assessing this capability. As detailed in Table 3, the assessment utilizes two metrics: MMR (Multi-modal Multi-step Reasoning) and CMD (Case Murder Detail), each with its distinct evaluation method. The MMR metric is primarily evaluated through four multiple-choice questions labeled in Section 3.2.3, scored similarly to perception questions. The CMD metric requires the agent to present its conclusions about the murderer's method and motive in

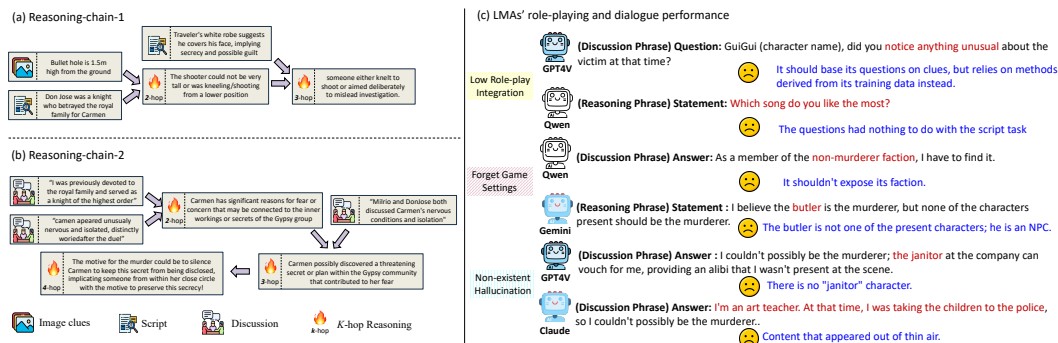

Figure 4: Qualitative analysis on WhodunitBench. Left: Reasoning chains generated by GPT-4V; Right: LMAs' role-playing and dialogue performance during games.

the form of open-ended responses. The answers provided by the agent will be compared with the ground truth outlined in Section 3.2.3 and automatically scored using GPT-4.

## 5.2 Evaluation Results

**Multimodal agents demonstrate suboptimal performance during discussion phases.** Table 3 shows that even the more capable GPT-4V only achieves an average score of approximately 20 during these segments. This finding suggests that the discussion phase provides limited assistance for all multimodal agents in fulfilling game-related tasks. This trend may be indicative of agents predominantly issuing irrelevant remarks, rather than delivering effective information conducive to reasoning and problem-solving. These observations underscore the need for improvements in how multimodal agents integrate into the gaming world and embody their roles, highlighting a significant gap in their ability to leverage discussion segments to facilitate game progression effectively.

**The CoT reasoning framework does not always bring benefits.** Although it can improve most performance indicators in Table 3, it sometimes reduces the effective output during the dialogue phase. As pointed out in some studies, murder mystery games are games of incomplete information, and advanced reasoning frameworks like CoT are not always guaranteed to be effective in such environments. Moreover, when dealing with tasks that do not require deep reasoning, such as directly recognizing text from images, introducing CoT might instead lead to a significant decline in performance. This emphasizes the need for precise adjustment based on the specific requirements of the task when selecting and applying reasoning techniques.

## 5.3 Further Analysis

**Qualitative analysis** We show a case of reasoning chains generated by GPT-4V in Figure 4(Left). We can observe that it reason over the available clues including (a) textual and visual clues obtained from character scripts; (b) useful dialogue information collected from other roles during the discussion phase.

There are some issues with LMAs in role-playing as shown in Figure 4(Right). It includes low role-play integration (i.e., not basing inquiries on existing clues within the game), forgetting the game settings, and hallucination (e.g., agents may refer to characters not mentioned in the script).

**Correlation between the CoE evaluation scores and the win rates in the online arena** By analyzing and comparing the metrics from Tables 2 and 3, it is evident that LMAs with higher scores in the CoE assessment also have higher win rates in the arena. *This suggests that the CoE evaluation method is effective in providing detailed insights into performance within competitive arenas.* Among the CoE metrics, reasoning-related metrics exhibit the strongest correlation with win rate, suggesting that reasoning capabilities are the most significant contributors to success.

# 6 Limitations and Potential Societal Impact

Our benchmark, WhodunitBench, features two modes: an online arena and a chain evaluation, designed to assess LMAs in realistic scenarios. This setup mirrors human behavior by requiring LMAs to combine multiple abilities at once, rather than isolating skills in controlled experiments. However, potential concerns and limitations remain regarding the evaluation methodology, metric design, and current data collection practices.

**Combination of Social Skills and Reasoning Abilities.** We find that the current evaluation intertwines interaction and reasoning, making the results less interpretable. In the data annotation process, we labeled critical clues that necessitate interaction to be uncovered. To separate interaction from reasoning, these key clues can be directly provided to agents in the "no-murderer" faction, enabling an analysis of each aspect's individual impact on scoring. Although this approach has been attempted, more effective solutions may exist for addressing this issue.

**The Kind of Reasoning Abilities.** Murder mystery games primarily assess core reasoning skills, such as logical deduction, visual-text detail verification, timeline reasoning and hypothesis testing. These games do not cover all reasoning abilities, particularly in computer programming. Strong performance in these games does not guarantee proficiency in all reasoning contexts. However, we believe that, the skills developed, like logical analysis and detail interpretation, are foundational and can be extended to other domains, holding significant potential for broader application.

**Cost considerations are pivotal.** Given that the script for each character often exceeds 5,000 words, the volume of data required to effectively test multimodal agents, such as GPT-4V, is substantial. Therefore, the evaluation on WhodunitBench is more expensive compared to existing benchmarks.

Additionally, we believe our benchmark has minimal societal impact. However, as agents integrate into daily life, the accuracy of our evaluations could shape public perception of their capabilities, possibly leading to unintended consequences.

# 7 Conclusion

In this work, we propose WhodunitBenchfor evaluating LMAs' capability in multi-modal perception, interaction, multi-step reasoning and goal execution. It includes 50 meticulously curated scripts and over 3000 closed-ended multiple-choice questions, along with corresponding open-ended queries featuring human-annotated ground truth. This framework supports online arena-style evaluations and enables detailed chain-linked assessments to evaluate specific capabilities at each stage of the game. Experiments demonstrate that existing LMAs struggle to perform complex tasks requiring compositional skills in dynamic interactive environments; even the state-of-the-art GPT-4V achieves a low score. We hope this work will guide future advancements, establishing a solid foundation for the continued development of LMAs.

## Acknowledgement

This work was supported in part by the National Natural Science Foundation of China (NO. 62322608), in part by the Guangxi Key R&D Project (No. AB24010167), the Project (No. 20232ABC03A25),in part by the Futian Healthcare Research Project (No.FTWS002), and in part by the Longgang District Special Funds for Science and Technology Innovation (No.LGKCSDPT2023002).

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

# A Appendix

## A.1 Preliminary: Murder Mystery Game

To better demonstrate our game process and testing procedures, we have created a **demo** and placed it in our open-source dataset and code repository.

### A.1.1 Game Introduction

In this section, we provide a detailed introduction to the rules and key elements of the murder mystery game, elucidating its suitability as an ideal dynamic environment platform for evaluating the multifaceted capabilities of LMAs. In the murder mystery game, each session is orchestrated through a meticulously designed script that constructs a self-contained fictional world. This world is enriched with elaborate backstories, sophisticated character development, and complex narrative structures. The script comprises the following primary elements, as shown in Figure 5:

- **Script.** Each murder mystery game unfolds according to a detailed script that establishes the game's universe and background. Within this setting, a unique murder mystery unfolds, featuring a core group of suspects, each equipped with their own **character script**. These scripts furnish extensive details about each character's name, faction, and backstory.
- **Role.** In real-life murder mystery games, each script is populated with multiple roles, each possessing a unique background and crucial clues that offer **diverse perspectives** on the script's virtual world. While any role could be suspected of the murder, only one is the true murderer. This divides the roles into two groups: **non-murder suspects** and the **murder suspect**.
- **Clue.** Clues are the pieces of information necessary to solve the murder mystery within the game, encompassing both the textual clues found in each character's script and the publicly shared visual clues. Each role receives different clues at the start of the game. Some roles will possess **key clues**. Additionally, certain clues may be misleading, requiring players to engage in deeper reasoning to fully understand their implications.

**Game Process:** We have simplified the entire game process into four stages, ① **Initialization phase** ② **Discussion phase** ③ **Reasoning phase** ④ **Voting phase**. In the **Initialization Phase** of the scripted murder mystery game, participants thoroughly examine their character scripts, which include essential details such as name, identity, interpersonal relationships, and pivotal text clues regarding the sequence of events on the day of the incident. Then they introduce their respective roles to start the game. The players are also given visual clues outside of their scripts needed to solve the puzzle. In the **Discussion Phase**, they engage in in-depth discussions, using both individual clues and shared evidence to analyze and share insights. During the **Reasoning phase**, each player needs to combine the direct clues they have obtained (including textual and visual clues) and the content of discussions to infer implicit clues. And then deduce the murderer and his modus operandi from this. The game culminates in the **Voting Phase**, where players, based on the evidence and discussions, decide on the murderer's identity.

Overall, the primary objective of these games is to decipher a murder mystery, which entails identifying the murderer, elucidating the method of the crime, and understanding the underlying motives. At the commencement of each game, players choose and embody specific roles, each equipped with a script that provides a unique vantage point and critical information pertinent to their role within this fabricated world. As the game progresses, players must analyze overt clues and engage in substantive discussions with fellow participants to amass information related to the crime. This ongoing process of information collection and inference through interactive collaboration enables players to gradually synthesize a comprehensive portrait of the case, uncovering the veritable truth.

### A.1.2 Action and Observation

**Observation space for each Phase.** The observation space varies across these stages, which will be detailed further. (1) **Initialization:** In this phase, the observation space for character-playing agents includes clues from role scripts and image clues, as well as introductions to other characters. This covers each character's background, identity, relationships, and the circumstances on the incident day, alongside public clues. Notably, observations can vary even within the same script, as agents have access to different information, leading to inconsistencies and incompleteness in the observed data. (2) **Discussion:** In this phase, the primary observation space for each role-playing agent includes

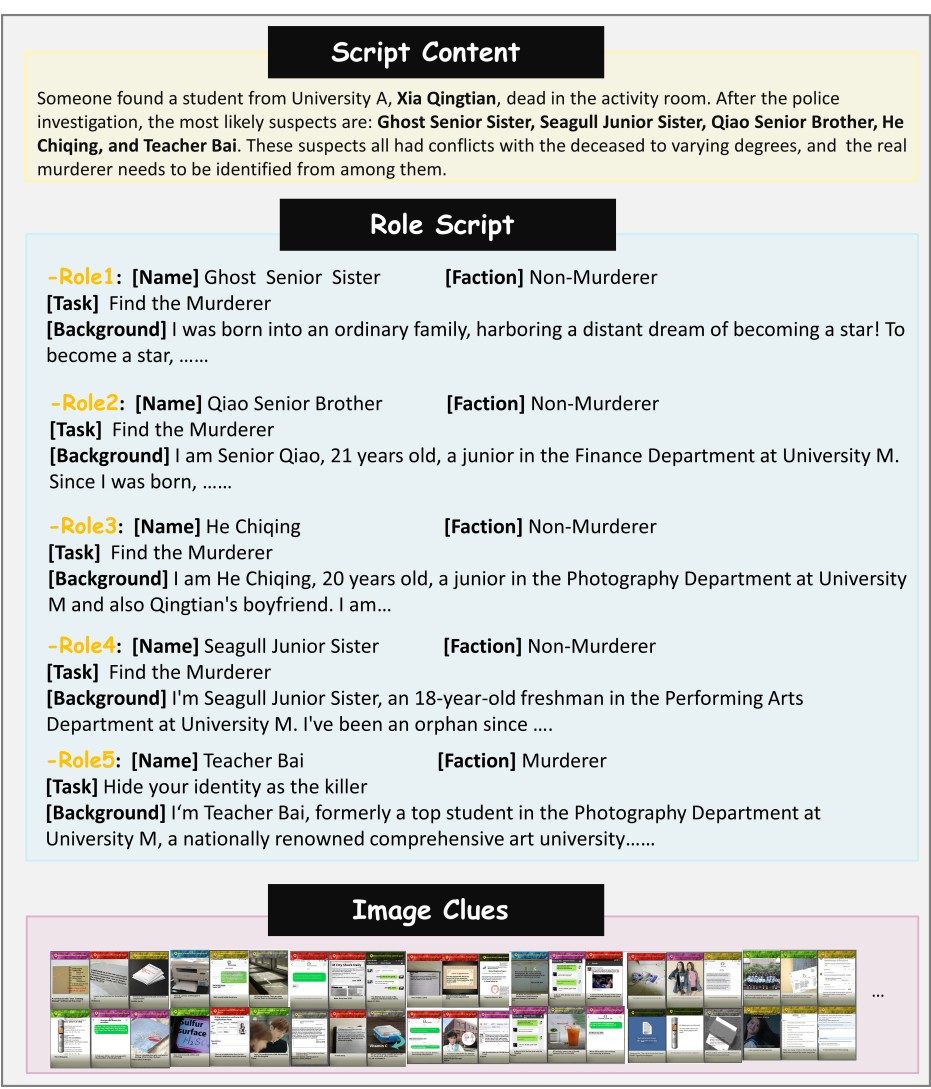

Figure 5: Illustration of primary elements in murder mystery games: scripts, roles and clues. Due to space limitations, the role scripts are detailed in Figure 6.

the statements and discussions among participants. This encompasses interpretations and inferences regarding the case facts, analyses of pictorial and textual clue cards, exchanges of questions and answers, interrogation and verification of case evidence, and debates over various deductive paths and conclusions. (3) **Reasoning:** In this phase, the observation space involves other roles discussing the current murderer's motive and method. Each role can refine or optimize their statements based on what others have said. (4) **Final Voting:** In this stage of the game, the observation space consists of other roles' voting. Each role can adjust their own response based on the votes of others.

**Action space for each Phase.** The action space varies across these stages, which will be detailed further. (1) **Initialization:** At this stage, the actions of the multimodal intelligent agents are based on the current observation to perform self-introduction. This includes not only conveying their basic information and functionalities but also demonstrating their understanding of the environment and how to navigate and interact within it effectively. (2) **Discussion:** At this stage, the multimodal intelligent agents should execute two actions: ① Share and analyze the clues in the script and discuss how these clues are connected to the current situation. ② Pose questions to other roles, which can be about clues related to that role or about suspicions concerning that role. (3) **Reasoning:** In this phase, the agent's action is to articulate their thoughts on the current suspect's motive and reasons for

committing the crime. (4) **Final Voting.** At this stage, there is only one action, which is to identify the **Murderer** and provide the motive and method of the crime.

## A.2 Additional Details on the Dataset

### A.2.1 Dataset Collection and Access

In our research, we introduce a benchmark based on murder mystery games, named WhodunitBench, designed to evaluate large multimodal agents (LMAs). We provide a detailed description of the dataset collection and verification processes as follows:

**Collection.** In this study, we invited five seasoned experts from the murder mystery game domain to select and extract suitable scripts. Each expert meticulously screened the provided script library based on detailed selection criteria, choosing the top 50 scripts that best met the standards, for which they received a compensation of $100 each. Furthermore, the author personally conducted a thorough review to ensure all selected scripts met the expected quality and thematic requirements. The primary resources we utilize in our work are the cloud services provided by the agent's company. Running all 50 scripts through a single model at once requires approximately 20 million tokens. We estimate the average hourly wage for the annotators to be approximately $8 per hour.

During the question annotation phase, we employed ten experienced scripted murder mystery game experts to annotate the data, with a compensation of $0.50 per question. Given the importance and complexity of constructing intricate reasoning chains, each reasoning chain was priced at $2 for annotation. To ensure the quality of annotations, we also hired three experts to review the questions at a cost of $0.20 per question. All data reviews were conducted strictly to ensure accuracy and reliability.

**Refinement and Verification.** To ensure the standardization and accuracy of our questions, we have established a meticulous proofreading and verification process. After the initial tagging is completed, each tagged question and reasoning chain undergoes a three-stage review to ensure it meets preset standards and logical correctness: ① Preliminary Review Stage: This stage is conducted by three experts who are separate from the tagging team. They are primarily responsible for checking the grammar, spelling, and format of the questions to ensure they meet predetermined standards and for confirming the basic logic and factual accuracy of the questions. This step is designed to ensure that all questions are clear and accurate before proceeding to a more in-depth logical review. ② Logic and Consistency Review: After passing the preliminary review, each question and reasoning chain enters the logic review stage. At this stage, the review team delves into the logical processes and structure within the questions and reasoning chains to ensure that each link is logically rigorous and closely connected to the overall plot of the script. In particular, reviewers check for logical gaps or leaps in reasoning. ③ Final Confirmation Stage: After rigorous reviews in the previous two stages, all questions and reasoning chains are submitted to the authors for final review. In this stage, in addition to reconfirming the accuracy and logic of the questions, the difficulty level is also assessed to ensure it is appropriate. Furthermore, the authors integrate all review comments to refine and optimize the questions, ensuring each one meets the highest quality standards.

### A.2.2 Additional Quantitative Examples

In this section, we provide additional examples of scripts used to configure the arena and questions for evaluation: (1) Further script examples are shown in Figure 7, 8. (2) Additional examples of question-answer (QA) are illustrated in Figure 9 and 10, which displays the multi-step reasoning questions we have marked, as well as the specific reasoning chains used during the marking process.

## A.3 Dataset Documentation

### A.3.1 Motivation

**For what purpose was the dataset created?** We created this dataset as a comprehensive test-bed for evaluating the perception, role-playing interaction, cognition and goal-achievement capabilities of large multimodal agents via murder mystery games.

### A.3.2 Composition

**What do the instances that comprise the dataset represent (e.g., documents, photos, people, countries)?** The instances in the dataset represent two main types of data. The first type includes elements for constructing the game environment of scripted murder mystery games, such as overall scripts, character scripts, and graphic and textual clues. The second type includes data for evaluating agents, comprising multiple-choice questions and open-ended questions.

**Does the dataset contain all possible instances or is it a sample (not necessarily random) of instances from a larger set?** Our script dataset is a sample from an existing collection of real-world scripted murder mystery game data, while the evaluation dataset consists of new annotations created by us.

**What data does each instance consist of?** In the first part of the script dataset, each instance consists of background script information, character script information, and graphic and textual clues within the script. In the second part, the evaluation dataset, each multiple-choice question instance includes a set of images, a long text passage, a question with options, and the correct answer. Each open-ended question instance includes a question, and the correct answer.

**Is there a label or target associated with each instance?** Yes, there is a label or target associated with each instance. In the script dataset, each instance includes roles, clues, and context information, which can be considered as targets or labels depending on the task. In the evaluation dataset, the multiple-choice questions and open-ended questions have correct answers that serve as the targets or labels for each instance.

**Is any information missing from individual instances?** No, there is no information missing from individual instances. Each instance in both the script dataset and the evaluation dataset is complete with all necessary information

**Are relationships between individual instances made explicit (e.g., users' movie ratings, social network links)?** No, the relationships between individual instances are not explicitly defined. Each instance in the script and evaluation datasets is treated independently without direct links to other instances.

**Are there recommended data splits (e.g., training, development/validation, testing)?** No, there are no recommended data splits. Users of the dataset can define their own splits based on their specific needs and goals.

**Are there any errors, sources of noise, or redundancies in the dataset?** Yes, there are potential sources of noise and redundancies in the dataset. Since the script dataset is sourced from existing open-source scripted murder mystery games, some scripts may contain inconsistencies or ambiguities that can introduce noise despite our best efforts to minimize them.

**Is the dataset self-contained, or does it link to or otherwise rely on external resources (e.g., websites, tweets, other datasets)?** The dataset is not entirely self-contained as it includes scripts sourced from real-world scripted murder mystery games. While these scripts are included in the dataset, they originate from publicly available resources. Despite this, all evaluation data and annotations are self-contained within the dataset and do not rely on external resources.

**Does the dataset contain data that might be considered confidential (e.g., data that is protected by legal privilege or by doctor–patient confidentiality, data that includes the content of individuals' non-public communications)?** No, the dataset does not contain any data that might be considered confidential. The scripts are sourced from publicly available scripted murder mystery games, and all evaluation data and annotations were created specifically for this dataset.

**Does the dataset contain data that, if viewed directly, might be offensive, insulting, threatening, or might otherwise cause anxiety?** The dataset does not intentionally contain any data that might be offensive, insulting, threatening, or anxiety-inducing. However, given the nature of scripted murder mystery games, some scripts may include themes of crime, violence, or other mature content that could be sensitive to some users. We have made efforts to review and filter the content to minimize potential issues. Additionally, if users identify any content they find problematic, we encourage them to report it to us, and we will take steps to replace or remove such content as necessary.

**Does the dataset identify any subpopulations (e.g., by age, gender)?** No

**Is it possible to identify individuals (i.e., one or more natural persons), either directly or indirectly (i.e., in combination with other data) from the dataset?** No, it is not possible to identify individuals, either directly or indirectly, from the dataset. The dataset consists of fictional characters and scenarios from scripted murder mystery games and does not include any real personal information or data that could be used to identify natural persons.

**Does the dataset contain data that might be considered sensitive in any way (e.g., data that reveals race or ethnic origins, sexual orientations, religious beliefs, political opinions or union memberships, or locations; financial or health data; biometric or genetic data; forms of government identification, such as social security numbers; criminal history)?** No. The dataset focuses solely on fictional characters and scenarios from scripted murder mystery games, without including any real or sensitive personal information.

### A.3.3 Collection Process

**How was the data associated with each instance acquired?** The data associated with each instance was acquired through two main methods. The script dataset was sourced from a collection of publicly available, real-world scripted murder mystery game scripts. The evaluation dataset, on the other hand, was created by our team, who carefully annotated and developed multiple-choice and open-ended questions based on the content of the scripts.

**What mechanisms or procedures were used to collect the data (e.g., hardware apparatuses or sensors, manual human curation, software programs, software APIs)?** The data was collected using a combination of manual curation and various software tools. For the game scripts, we downloaded them from publicly available online resources. We then used OCR (Optical Character Recognition) technology to process and extract the script content. The images within the scripts were processed using image cropping techniques to obtain the clue information. The evaluation dataset was created by our team. We manually annotated the questions and correct answers based on the script content. For generating distractor options for the multiple-choice questions, we used GPT-4, and our team further refined these options to ensure quality and relevance. The entire process involved meticulous manual work complemented by advanced software tools to ensure accuracy and consistency.

**If the dataset is a sample from a larger set, what was the sampling strategy (e.g., deterministic, probabilistic with specific sampling probabilities)?** The dataset is a sample from a larger set, and the sampling strategy was deterministic. We enlisted the expertise of seasoned murder mystery game experts to ensure quality and applicability. The scripts were sourced from industry-recognized creative teams and platforms, with selection criteria focusing on: ① Scientific Integrity: Excluding scripts with supernatural phenomena to ensure realistic resolutions. ② Content Complexity: Choosing scripts with high reasoning complexity to test deductive capabilities. ③ Logical Coherence: Ensuring logical soundness with balanced evidence and clues.

**Who was involved in the data collection process (e.g., students, crowdworkers, contractors) and how were they compensated (e.g., how much were crowdworkers paid)?** In this study, five seasoned experts from the murder mystery game domain were invited to select and extract suitable scripts. Each expert screened the script library and chose the top 50 scripts that best met our criteria, receiving a compensation of $100 each. The author personally reviewed all selected scripts to ensure quality and thematic alignment. For the question annotation phase, we employed ten experienced murder mystery game experts. They were compensated at $0.50 per question for general annotations and $2 per intricate reasoning chain. Additionally, three experts were hired to review the annotations at $0.20 per question to ensure accuracy and reliability.

**Over what timeframe was the data collected?** The data was collected over a period of three months, from March 2024 to May 2024. During this time, the script selection, annotation, and review processes were conducted to ensure the quality and applicability of the dataset.

**Were any ethical review processes conducted (e.g., by an institutional review board)?** No

**Did you collect the data from the individuals in question directly, or obtain it via third parties or other sources (e.g., websites)?** We did not collect the data directly from individuals. The script data was obtained from publicly available sources on websites, curated and selected by seasoned murder mystery game experts. The evaluation data, including annotations and questions, was created by our team of experts based on the sourced scripts.

**Were the individuals in question notified about the data collection?** N/A

**Did the individuals in question consent to the collection and use of their data?** N/A

**If consent was obtained, were the consenting individuals provided with a mechanism to revoke their consent in the future or for certain uses?** N/A

**Has an analysis of the potential impact of the dataset and its use on data subjects (e.g., a data protection impact analysis) been conducted?** No

### A.3.4 Preprocessing/cleaning/labeling

**Was any preprocessing/cleaning/labeling of the data done (e.g., discretization or bucketing, tokenization, part-of-speech tagging, SIFT feature extraction, removal of instances, processing of missing values)?** Yes, extensive preprocessing, cleaning, and labeling of the data were performed. For the script dataset, we used OCR technology to extract text from images and processed the images for clarity. We also performed manual cleaning to remove any inconsistencies and ensure logical coherence. For the evaluation dataset, we annotated questions and correct answers, and generated distractor options using GPT-4, followed by manual refinement.

**Was the "raw" data saved in addition to the preprocessed/cleaned/labeled data (e.g., to support unanticipated future uses)?** Yes, the raw data was saved alongside the preprocessed, cleaned, and labeled data to support unanticipated future uses and to ensure transparency and reproducibility.

**Is the software that was used to preprocess/clean/label the data available?** No

### A.3.5 Uses

**Has the dataset been used for any tasks already?** No, this dataset has not been utilized for any tasks before the baseline experiments conducted in this paper.

**Is there a repository that links to any or all papers or systems that use the dataset?** No

**What (other) tasks could the dataset be used for?** The dataset can be used to test multimodal agents' abilities to perceive, reason, and make decisions in dynamic, incomplete information environments. It aims to assess how well agents can complete tasks in a manner akin to human behavior, addressing the significant challenge of developing a theory of mind to navigate complex scenarios.

**Is there anything about the composition of the dataset or the way it was collected and preprocessed/cleaned/labeled that might impact future uses?** We will continue to maintain the dataset and attempt to expand its scale to achieve a more comprehensive evaluation.

**Are there tasks for which the dataset should not be used?** Yes, the dataset should not be used for tasks that require large-scale training data due to its limited size.

### A.3.6 Distribution

**Will the dataset be distributed to third parties outside of the entity (e.g., company, institution, organization) on behalf of which the dataset was created?** The dataset is open-source, and we will also provide the scripts we used. However, it is important to note that we do not claim any rights over the scripts.

**How will the dataset will be distributed (e.g., tarball on website, API, GitHub)?** The dataset will be distributed via a GitHub repository.

**When will the dataset be distributed?** The dataset will be made open-source after a final review by our team.

**Will the dataset be distributed under a copyright or other intellectual property (IP) license, and/or under applicable terms of use (ToU)?** No

**Have any third parties imposed IP-based or other restrictions on the data associated with the instances?** No

**Do any export controls or other regulatory restrictions apply to the dataset or to individual instances?** No

### A.3.7 Maintenance

**Who will be supporting/hosting/maintaining the dataset?** All the authors.

**How can the owner/curator/manager of the dataset be contacted (e.g., email address)?** Contact by email at any time

**Is there an erratum?** No

**Will the dataset be updated (e.g., to correct labeling errors, add new instances, delete instances)?** It may be updated, and if necessary, we will propose modifications on our GitHub.

**If the dataset relates to people, are there applicable limits on the retention of the data associated with the instances (e.g., were the individuals in question told that their data would be retained for a fixed period of time and then deleted)?** No

**Will older versions of the dataset continue to be supported/hosted/maintained?** No

**If others want to extend/augment/build on/contribute to the dataset, is there a mechanism for them to do so?** Yes, if others wish to extend, augment, build on, or contribute to the dataset, they can do so by submitting pull requests or opening issues on our GitHub repository. We encourage community contributions and aim to review and integrate them in a timely manner to enhance the dataset.

### A.4 Additional Details on the Evaluation Benchmark

### A.4.1 Prompts

To enable LMAs to perform within our WhodunitBench, we introduce a series of structured prompts. The categories of prompt templates we use are detailed in the table 4. The specific content for each prompt type is presented in Figures 11, 12, 13 and 14. The *symbol* attribute within the table links directly to the corresponding detailed contents.

Table 4: Detailed description of the prompt

| Category | Name | Symbol | Description |
|---|---|---|---|
| **System** | Rules | $I_e$ | Describes the rules and procedures of the Game |
| | Script | $I_S$ | Provide the agent with its role details |
| | Live-info | $I_d$ | Real-time information about the current game, such as dialog information |
| **Action** | Introduction | $I_i$ | The action that involves asking the agent to introduce itself |
| | Discussion | $I_D$ | The action that prompts the agent to discuss and choose an action |
| | Reasoning | $I_r$ | The action that directs the agent to identify the murderer and their motive |
| | Voting | $I_v$ | The action that directs the agent to vote the murderer |
| **Evaluation** | RP | $I_P$ | Prompts used to evaluate the naturalness of agent role-playing |
| | SPC | $I_p$ | Prompts used to evaluate the degree of agent role immersion |
| | CMD | $I_c$ | Prompts used to score the agent's final reasoning on the motive and method of the crime |

### A.4.2 Additional Examples of Dialogue Content

Figures 15, 16 and 17 display examples of dialogue content generated by LMAs at various stages of the game.

### A.4.3 Human Performance

Human performance serves as an upper bound for our benchmark. To obtain more rigorous and robust results, we plan to include a wider range of participants with diverse skill levels in future evaluations. And we have developed an interface that allows human participants to directly engage with different LMAs within a murder mystery game scenario. This setup not only offers participants a tangible sense of the differences between LMAs but also furnishes data that facilitates an in-depth analysis of human and agent behavior patterns, decision-making processes, and the efficacy of human-agent collaboration. These insights are invaluable for the continued development of intelligent systems.

## A.5 Author Statement

The scripts used in this study were collected from publicly available online websites. All scripts were gathered within the scope of public accessibility, ensuring compliance with relevant data usage and privacy policies. We acknowledge that all intellectual property rights of the collected scripts belong to the original authors or platforms, and we thank them for creating and sharing these resources. These resources are used solely for academic research, and we pledge not to use this data for any purposes unrelated to research. The annotated data is marked by our team, and we own the copyright.

| Script: Campus Belle Death Incident | |
|---|---|
| **Role Name** | **Role Script** |
| Teacher Bai | I am **Teacher Bai**, formerly a top student in the Photography Department at University M, which is a nationally famous comprehensive arts university. The Photography Department there is especially well-known. In terms of photography skills, I believe no one at University M can surpass me. A year ago, I graduated with outstanding results as a postgraduate and stayed on to teach. Now, I am also the photography instructor for the anime costume club.[Appointment Letter - Anime Costume Club Photography Instructor: Bai Laoshi]Every school has its peculiar legends, and there's a legend at our university that the school beauty is cursed. Every two years, one dies. Four years ago, the school beauty fell to her death from the activities building. Two years ago, another school beauty fell from a building, causing widespread panic as everyone thought it was the curse. The one who fell two years ago was my girlfriend at the time- Xiaohua.........I found out that **she has a heart condition** and often takes medication, always carrying nitroglycerin with her. Thus, **a plan emerged**......
**On the day of the incident:** At 13:36, I hatched a plan and sent a text message to Ou Xuemei, apologizing for my earlier attitude and asking her to meet me at the rooftop of the school's activity center, a known rendezvous spot, at 14:00. I then sent another message to Xia Qingtian, wanting to discuss and resolve the rumors about us.[Text from Bai to Ou: "Sweetie, I was wrong earlier. Come to the rooftop at 14:00, and I'll apologize."] [Text from Bai to Xia: "Qingtian, I also think these rumors are too much and are damaging our reputations. Let's meet at the rooftop at 14:00 to discuss a solution."] At 13:55, I hid in a blind spot on the rooftop..... |
| He Chiqing | I am **He Chiqing**, a 20-year-old junior in the Photography Department at University M and also the boyfriend of Qingtian. I prefer a quiet life and love reading books, especially detective novels. Three years ago, on the first day of my freshman year in September 2013, I met Qingtian. She was so beautiful that I instantly fell in love with her. She was a classmate of mine, and we saw each other every day. I often sought her out to chat. For her, I later joined the "Anime Costume Club." On October 29, 2013, a little over a month after we started school, on Qingtian's birthday, I confessed my feelings to her! Qingtian happily agreed to date me. Soon after, we moved out of the dormitory and rented a house next to the school to live together. Since then, I was almost always by Qingtian's side. Qingtian loves photography and participates in various shooting activities; while I, enjoying writing, spent my time apart from joining her in various school activities writing detective stories at home. I feel my university life is simply perfect![A page from a calendar tucked in a book, dated October 29, 2013, with handwriting on it: Thank the heavens, thank the earth, thank the sunshine for letting us meet.]........
**On the day of the incident:** At 1:45 PM, I arrived at Room 601, handed the drink to Qingtian, and kept apologizing. She said she had encountered too many things recently and wanted to be alone for a while, asking me to leave. At 1:50 PM, I put down the juice, left the activity building, and went back to my house. At 3:00 PM, as I was anxiously waiting at home, I was summoned by a detective, who informed me that Qingtian had fallen from a building and committed suicide. |
| Ghost Senior Sister | I was born into an ordinary family, harboring a distant dream—I wanted to become a star! To achieve stardom, one must have a detailed plan, accomplishing set goals at each stage. In 2012, I was admitted to the Music Department of University M, a renowned institution for the arts. I could feel that I was one step closer to becoming a "star." With my good looks, lively and extroverted personality, and singing talent, I quickly rose to fame within the school and, as I wished, became the center of attention. [Star Cultivation Handbook Step One: Reject mediocrity! Become famous in school, Step Two: Become the campus belle, Step Three: Transform from campus belle to internet-famous campus belle.] However, the path to chasing dreams is never smooth; there are always some annoying stumbling blocks. In 2014, a girl from the photography department—Xiaohua—stole the glory that was rightfully mine. Her emergence turned me into just another ordinary "passerby" in the school. Every time I think about this, I gnash my teeth in frustration. In February 2014, the drama club posted a recruitment notice for the lead actress in the school anniversary play scheduled for March 15.......
**On the day of the incident:** At 14:30, I reached Classroom 601 on the 6th floor of the activity building and found Xia Qingtian lying on the desk, seemingly asleep. I quietly approached and covered her mouth with a handkerchief. After a few seconds, I nudged her, and she didn't respond. Taking advantage of her unconscious state, I took the evidence photos she had secretly taken from her bag. [In her possession, there was a photo dated 2014 showing "Ghost Sister" bullying Xiaohua.] I stuffed the photo and the handkerchief back into my bag, then walked over to the window to make sure no one was around the activity building before pushing Xia Qingtian out of the window.
By 14:40, I had left Classroom 601 having dealt with this major trouble, and I decided to go buy some snacks..... |
| Seagull Junior Sister | I am Seagull Junior Sister, an 18-year-old freshman at University M's Performing Arts Department. I've been an orphan since childhood and do not know my biological parents. However, I'm very content with my current family. My adoptive father, who has always doted on me like a princess, is a highly successful businessman owning the largest film and television company in the country, HE BEAÚTY. He is refined and progressive, and has taught me many life lessons, which I deeply respect. [Clue: a warm and affectionate photo with my adoptive father] In September 2015, I was thrilled to be admitted to University M. At the anime club's recruitment event, I met my honey—Bai Laoshi. He is an outstanding teacher at our school, with a smile as warm as the spring breeze and wisdom as dazzling as the stars, fulfilling all my girlish fantasies. However, I was not selected to join the anime club, and I thought I would just continue to silently adore him. On March 2, 2016, he secretly found me on the rooftop and confessed that he had fallen for me at first sight and asked me to be his girlfriend. How could I possibly refuse such a confession?.......
**On the day of the incident:** At 2:02 PM, I saw Xia Qingtian; she saw me and ran downstairs in a panic. Was she nervous, or was she just stunned by how good I looked? I waited on the rooftop for Bai Laoshi for over ten minutes, but he didn't show up. I texted him asking why he didn't come, and he replied that something had come up and he would apologize another day. At 2:15 PM, I returned to my dorm, annoyed, and watched an idol drama. At 3:00 PM, I was informed that Xia Qingtian had fallen from a building. Since I had been in and out of the activity building, I was called in for questioning...... |
| Qiao Senior Brother | I am Qiao Xuechang, a 21-year-old finance major in my third year at University M. From the moment I was born, I was the designated heir of the Qiao family enterprise due to my mother's frail health, which allowed her to have only one child. My father, mother, and even our household staff told me I was the sole inheritor of our family business. However, when I was 10 years old (2005), shortly after my mother passed away, my father unexpectedly brought home a boy one year older than me! That was when I learned for the first time that my father had an illegitimate son. My father instructed me to call this boy my older brother. This "brother" excelled in academics and his demeanor was so similar to my father's that my father doted on him, frequently remarking that he resembled his younger self. Thus, this illegitimate brother shared my home, my father, and even my inheritance rights! [Photo: Dad, the illegitimate son, and I, with the illegitimate son's face marked with a censor mark.]My father announced that he would choose the most suitable between us two to inherit the company. In 2010, I inherited my mother's artistic talents but struggled academically, achieving poor grades that only money could rectify, allowing me entrance into the best high school in M City. [Various artistic trophies—painting, music, calligraphy, essay competitions, etc.]In June 2012, my illegitimate brother was admitted to University M, a fact that my father celebrated grandly, even introducing him to our business contacts and involving him in our family company. In September 2012, I began my final year of high school, adopting a strenuous routine—sleeping at 2 AM and waking up at 5 AM to study, leading to slight improvements in my grades. [A study schedule hidden in an old notebook: 5:00-6:00 AM English, 6:00-7:00 AM ancient poetry, daytime classes, and night-time studies in mathematics, physics, chemistry, and biology from 10:00 PM to 2:00 AM.] Despite not being academically inclined, I was determined not to let what rightfully belonged to me fall into the hands of an illegitimate child. I had to get into University M to regain my father's trust and my inheritance![From childhood to high school, my academic transcripts showed poor performance, with a pre-university mock exam score of 276.] On June 1, 2013, I spent 200,000 to bribe the top student in my class—Xia Qingtian—to help me cheat in the college entrance exam. Despite her poor family background, Xia Qingtian agreed to assist me in exchange for the opportunity to attend university, which she also passed and attended University M, although we rarely interacted. [High school graduation photo with the whole class, including me and Xia Qingtian.]**On the day of the incident:** (On March 28, 2016) At 1:00 PM, I made a phone call to Xia Qingtian, pretending that I wanted to surrender myself and hoping to talk to her one last time. I asked her to meet me at 2:30 PM on the rooftop of the school's activity building on the eighth floor. [Cell phone call record; March 28, 2016, 1:00 PM, dialed Xia Qingtian, call duration 1 minute.] At 1:30 PM, using my computer, I forged and printed a suicide note supposedly signed by Xia Qingtian. [There is a printer and A4 paper in my room.] At 2:30 PM, I brought an ashtray to the rooftop of the activity building and placed the forged suicide note under a brick. I planned to knock Xia Qingtian unconscious and then throw her from the rooftop, creating the illusion of a suicide jump. [There is an ashtray in my backpack.] From 2:30 PM until an unknown time, I waited for Xia Qingtian, but she did not show up. At 3:00 PM, to my shock, Xia Qingtian fell from the building, landing on the lawn right below the rooftop! What happened? In a panic, I tried to leave the building but was stopped by the security guard, Uncle Sa. |

Figure 6: Detailed role scripts of Figure 5

## Script Content

On May 16, 2020, during a private party held at his residence "Ban Shan Villa," tycoon Hong Qing was abducted by a mysterious kidnapper who also detained the guests. When the police arrived, they found six guests and one corpse inside the villa. These guests were subsequently listed as suspects, but they insisted that they were also victims. Despite this, the truth behind the kidnapping and murder remains elusive, with all guests claiming no involvement in the case.

## Role Script

**-Role1**: **[Name]** Andy        **[Faction]** Non-Murderer
**[Task]** Find the Murderer
**[Background]** Some memories are gradually waking up in my mind. I remember that today is May 16, 2020. I am invited to attend a private party at Mr. Hong's Ban Shan Villa., ……

**-Role2**: **[Name]** Chen Xin        **[Faction]** Non-Murderer
**[Task]** Find the Murderer
**[Background]** I am a bodyguard named Chen Xin, and my boss is Hong Qing, who is commonly referred to as Mr. Hong. Today, May 16, 2020, ……

**-Role3**: **[Name]** Franklin        **[Faction]** Non-Murderer
**[Task]** Find the Murderer
**[Background]** On the appointed day, Chen arranged for a taxi to pick me up from the hotel and take me to a suburban mountainous area.…

**-Role4**: **[Name]** Red Snow        **[Faction]** Non-Murderer
**[Task]** Find the Murderer
**[Background]** On May 16, 2020, at the invitation of Mr. Hong, I went to his mountain villa for a private party. Mr. Hong is my business partner, and .….

**-Role5**: **[Name]** Vivian        **[Faction]** Murderer
**[Task]** Hide your identity as the killer
**[Background]** As a young live streamer, you and your boss Andy attend a mysterious gathering. Along the way, you encounter various characters, including Hongxue……

**-Role6**: **[Name]** Richard        **[Faction]** Non-Murderer
**[Task]** Find the Murderer
**[Background]** Richard, 40 years old, is the captain of the city's criminal investigation team. He grew up as an orphan, lacking parental care, and was bullied by his peers during his childhood. At the age of 10,……

## Image Clues

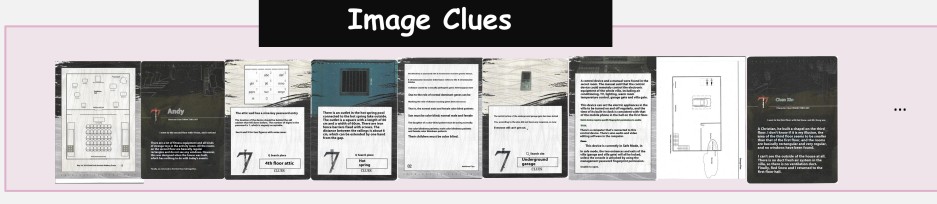

Figure 7: Additional examples of murder mystery game scripts utilized in our dataset.

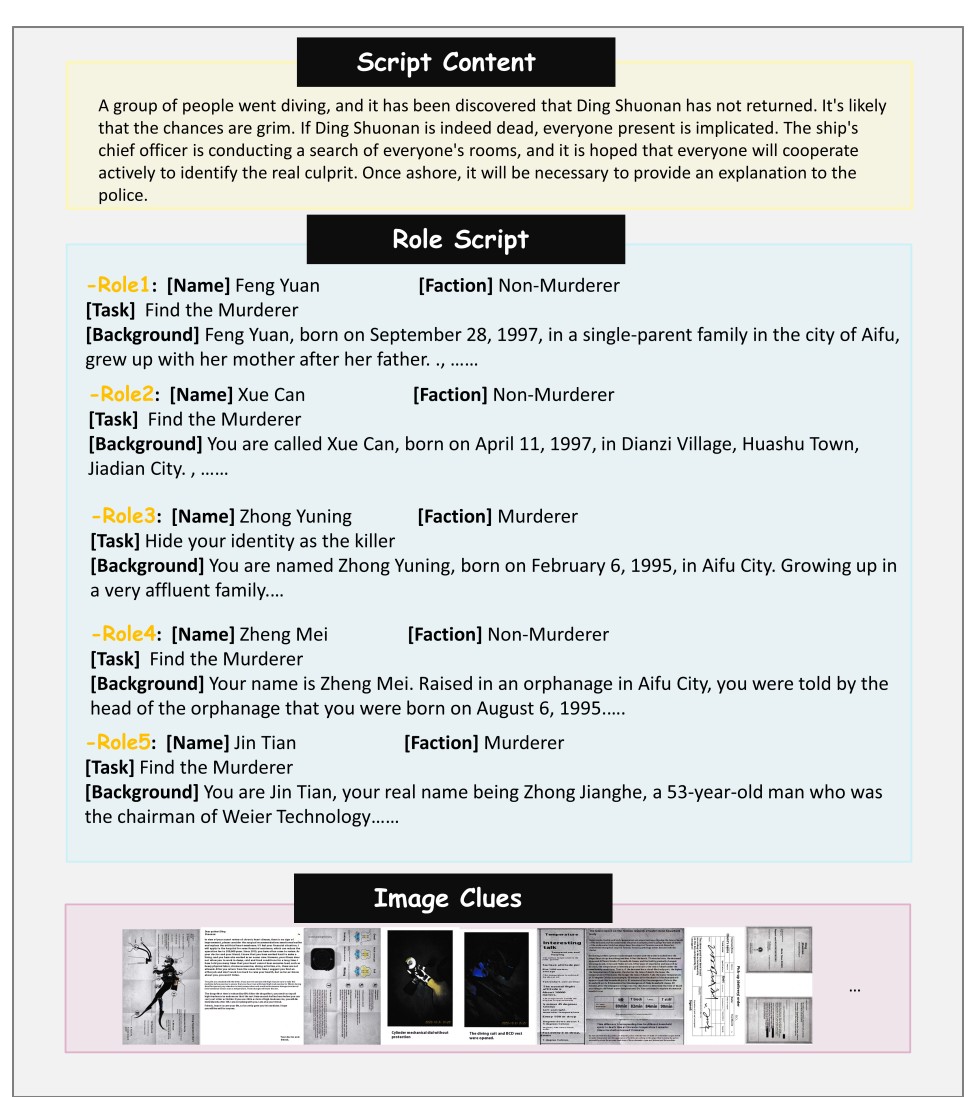

**Script Content**

A group of people went diving, and it has been discovered that Ding Shuonan has not returned. It's likely that the chances are grim. If Ding Shuonan is indeed dead, everyone present is implicated. The ship's chief officer is conducting a search of everyone's rooms, and it is hoped that everyone will cooperate actively to identify the real culprit. Once ashore, it will be necessary to provide an explanation to the police.

**Role Script**

**-Role1**: **[Name]** Feng Yuan          **[Faction]** Non-Murderer
**[Task]** Find the Murderer
**[Background]** Feng Yuan, born on September 28, 1997, in a single-parent family in the city of Aifu, grew up with her mother after her father. ., ……

**-Role2**: **[Name]** Xue Can          **[Faction]** Non-Murderer
**[Task]** Find the Murderer
**[Background]** You are called Xue Can, born on April 11, 1997, in Dianzi Village, Huashu Town, Jiadian City. , ……

**-Role3**: **[Name]** Zhong Yuning          **[Faction]** Murderer
**[Task]** Hide your identity as the killer
**[Background]** You are named Zhong Yuning, born on February 6, 1995, in Aifu City. Growing up in a very affluent family.…

**-Role4**: **[Name]** Zheng Mei          **[Faction]** Non-Murderer
**[Task]** Find the Murderer
**[Background]** Your name is Zheng Mei. Raised in an orphanage in Aifu City, you were told by the head of the orphanage that you were born on August 6, 1995.…..

**-Role5**: **[Name]** Jin Tian          **[Faction]** Murderer
**[Task]** Find the Murderer
**[Background]** You are Jin Tian, your real name being Zhong Jianghe, a 53-year-old man who was the chairman of Weier Technology……

**Image Clues**

Figure 8: Additional examples of murder mystery game scripts utilized in our dataset.

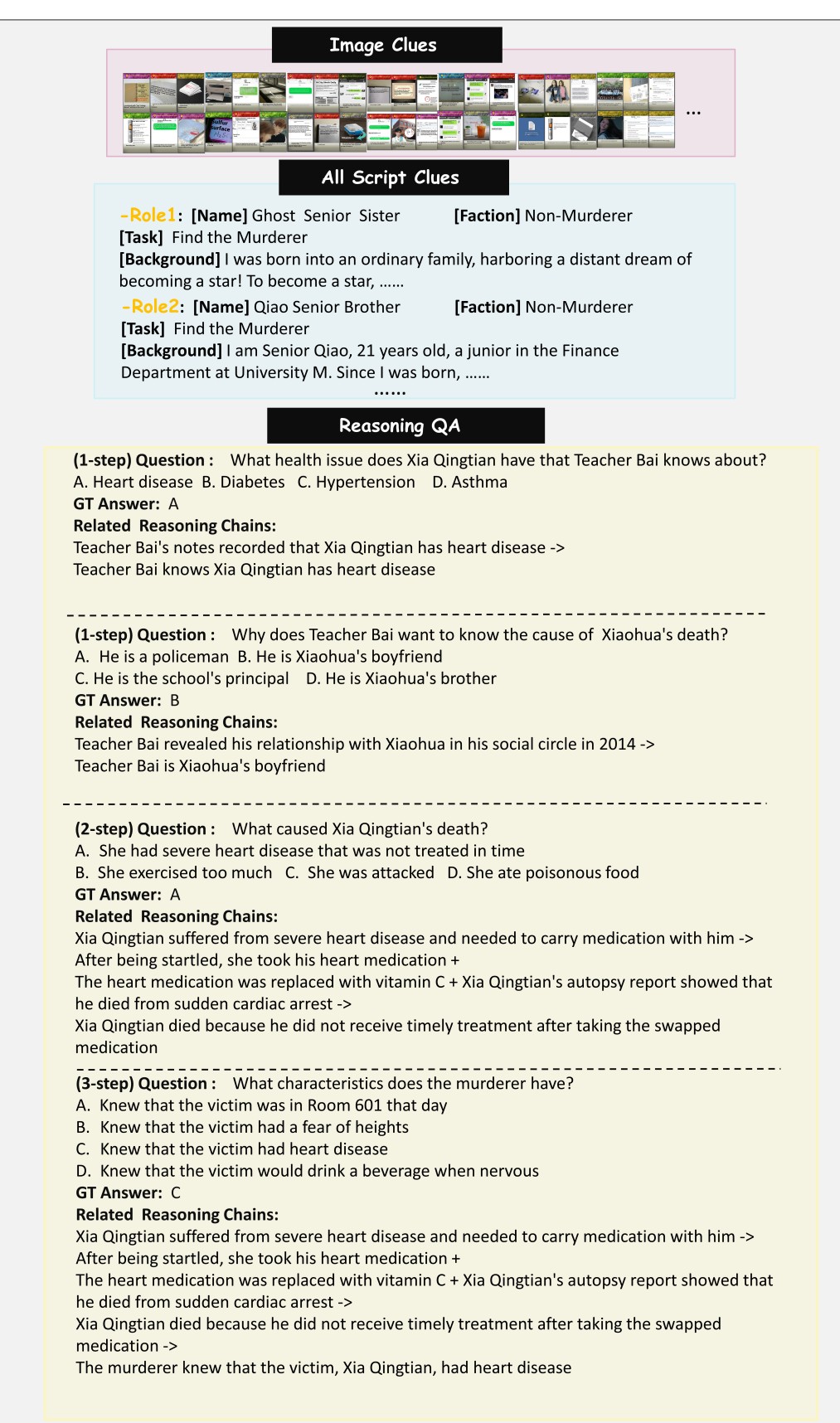

Figure 9: Additional examples of multi-step reasoning QA and corresponding reasoning chains.

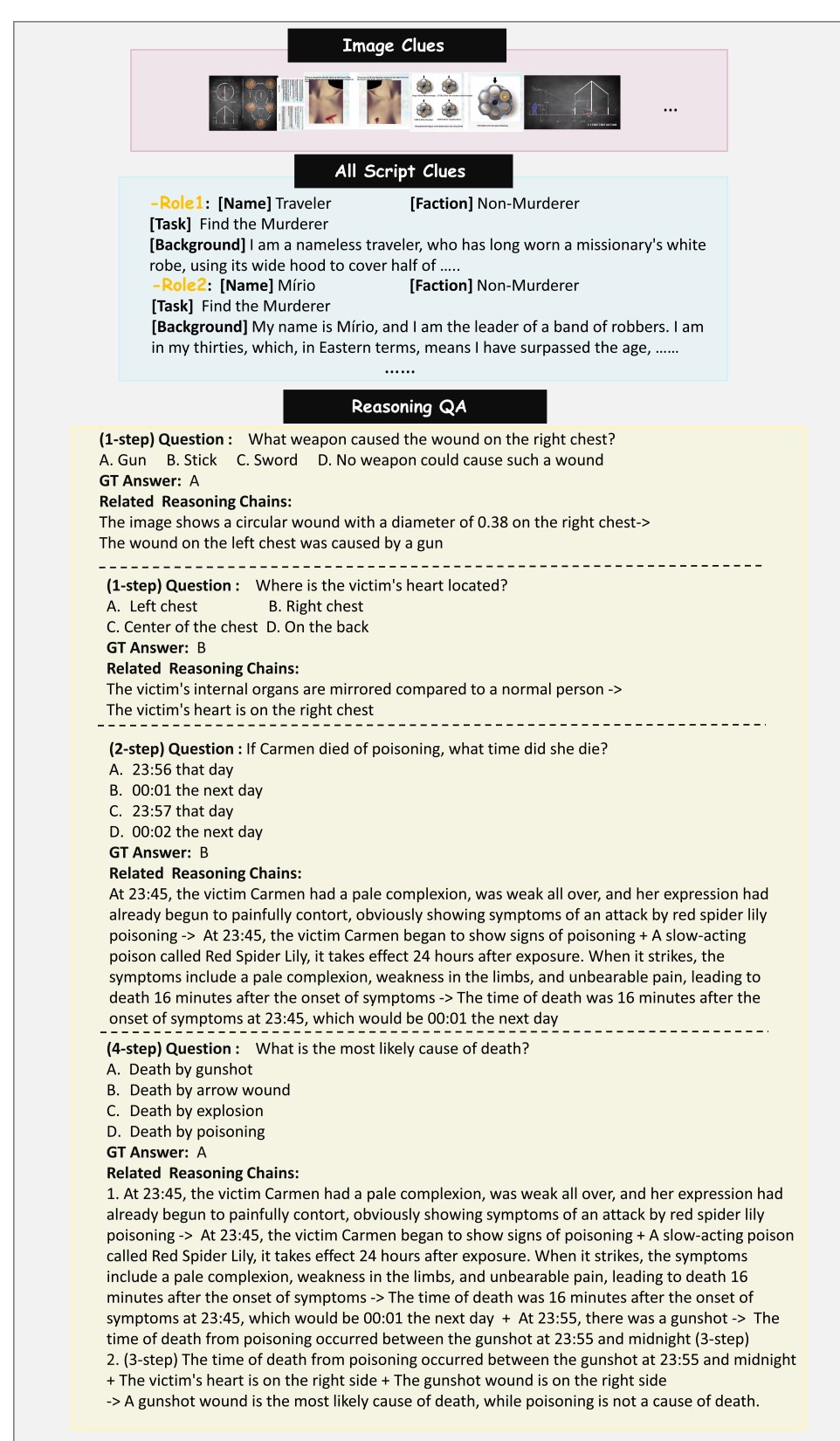

**Image Clues**

**All Script Clues**

-**Role1**: **[Name]** Traveler          **[Faction]** Non-Murderer
**[Task]** Find the Murderer
**[Background]** I am a nameless traveler, who has long worn a missionary's white robe, using its wide hood to cover half of …..

-**Role2**: **[Name]** Mírio          **[Faction]** Non-Murderer
**[Task]** Find the Murderer
**[Background]** My name is Mírio, and I am the leader of a band of robbers. I am in my thirties, which, in Eastern terms, means I have surpassed the age, ……

......

**Reasoning QA**

**(1-step) Question :**   What weapon caused the wound on the right chest?
A. Gun     B. Stick     C. Sword     D. No weapon could cause such a wound
**GT Answer:** A
**Related  Reasoning Chains:**
The image shows a circular wound with a diameter of 0.38 on the right chest->
The wound on the left chest was caused by a gun

- - - - - - - - - - - - - - - - - - - - - - - - - - - - - - - - - - - - - - - - - - - - - - - - -

**(1-step) Question :**   Where is the victim's heart located?
A.  Left chest           B. Right chest
C. Center of the chest  D. On the back
**GT Answer:** B
**Related  Reasoning Chains:**
The victim's internal organs are mirrored compared to a normal person ->
The victim's heart is on the right chest

- - - - - - - - - - - - - - - - - - - - - - - - - - - - - - - - - - - - - - - - - - - - - - - - -

**(2-step) Question :** If Carmen died of poisoning, what time did she die?
A.  23:56 that day
B.  00:01 the next day
C.  23:57 that day
D.  00:02 the next day
**GT Answer:** B
**Related  Reasoning Chains:**
At 23:45, the victim Carmen had a pale complexion, was weak all over, and her expression had already begun to painfully contort, obviously showing symptoms of an attack by red spider lily poisoning -> At 23:45, the victim Carmen began to show signs of poisoning + A slow-acting poison called Red Spider Lily, it takes effect 24 hours after exposure. When it strikes, the symptoms include a pale complexion, weakness in the limbs, and unbearable pain, leading to death 16 minutes after the onset of symptoms -> The time of death was 16 minutes after the onset of symptoms at 23:45, which would be 00:01 the next day

- - - - - - - - - - - - - - - - - - - - - - - - - - - - - - - - - - - - - - - - - - - - - - - - -

**(4-step) Question :**   What is the most likely cause of death?
A.  Death by gunshot
B.  Death by arrow wound
C.  Death by explosion
D.  Death by poisoning
**GT Answer:** A
**Related  Reasoning Chains:**
1. At 23:45, the victim Carmen had a pale complexion, was weak all over, and her expression had already begun to painfully contort, obviously showing symptoms of an attack by red spider lily poisoning -> At 23:45, the victim Carmen began to show signs of poisoning + A slow-acting poison called Red Spider Lily, it takes effect 24 hours after exposure. When it strikes, the symptoms include a pale complexion, weakness in the limbs, and unbearable pain, leading to death 16 minutes after the onset of symptoms -> The time of death was 16 minutes after the onset of symptoms at 23:45, which would be 00:01 the next day  + At 23:55, there was a gunshot -> The time of death from poisoning occurred between the gunshot at 23:55 and midnight (3-step)
2. (3-step) The time of death from poisoning occurred between the gunshot at 23:55 and midnight + The victim's heart is on the right side + The gunshot wound is on the right side
-> A gunshot wound is the most likely cause of death, while poisoning is not a cause of death.

Figure 10: Additional examples of multi-step reasoning QA and corresponding reasoning chains.

## Game Rules [System]

You are an expert in playing the social deduction game named " Murder Mystery " with some other players.
**\*Game Rules\*:**
1.Game Setup: Players receive scripts with background, faction (murder suspects or non-murder suspects), and missions at the start.
2.Objectives: Murder suspects aim to conceal their identity and mislead others, while non-murder suspects work to identify the murderer through clues and deduction.
3.Gameplay Dynamics: Murder suspects create false leads and accuse innocents, whereas non-murder suspects analyze clues from various sources to solve the mystery.
4.Conclusion: The game ends with players voting on who they suspect is the murderer, based on the motive and method deduced from the evidence.
5.Clue Integration: All necessary clues are provided to non-murder suspects, requiring them to integrate these with their knowledge to pinpoint the lone murderer in the murder suspect faction.
6.Among those present, one is definitely the murderer, who may lie or mislead others. Focus on the clues and discern each character's alignment during discussions to assess the truth of their statements.
7.If you are in the murderer's faction, try to hide anything that might expose you as the murderer or reveal your method of committing the crime.
8.During the discussion phase, the murderer can lie to lead others to incorrect conclusions.
9.If you are not the murderer, clear your suspicions, assure others of your innocence, and actively seek clues to identify the murderer. These clues may be graphical, textual, or hidden in each player's script. Engage actively with others during the discussion phase.
10. Only rely on evidence at the scene to prove innocence; no assumptions about other non-existent evidence are allowed.
11. Never reveal your faction at any time.
**\*Game Phases\*:**
1.Introduction Phase: Players introduce themselves using their script details, showcasing their character's identity and backstory to highlight distinct traits. Players receive a mix of text and visual clues related to the case, which require careful analysis to uncover hidden links or information.
2.Discussion and Interaction Phase: This phase involves discussions and interactions where players exchange clues, voice suspicions, and potentially form alliances or deceive others.
3.Accusation and Reasoning Phase: Players accuse someone of being the murderer based on their deductions, explaining their reasoning and the suspected motive and method. A majority vote determines the accused.

Figure 11: System prompts for game rule introduction.

## Role Script[System]

Next, I will give you all the information you have in this game:
Your character's name is {}
Your Background and Experiences:
{}
Your faction: {}
Your mission: {}
Now, the text clues is :
{}

## Live-Information[System]

Now, the dialog history is:
{}

Figure 12: System prompts for role scripts and live information introduction.

### Introduction[Actions]

Starting now, let's proceed with the first stage of self-introduction.
Please carefully read your  information and tips. Don't output irrelevant content.
Start by introducing yourself:

### Discussion [Actions]

All current conversations are as follows:
{}

It's time for the third stage of the discussion, where you need to do the following:
1. Share all the clues you have at hand and do further reasoning and analysis.
Here are some streamlined strategies you can use:
(1) If you belong to the non-murderer suspect faction, you are definitely not the murderer.
(2) If you are the murderer, shift attention towards others and mislead their reasoning. Conceal any clues in your script that point to you.
2. Choose one of the following characters to ask questions:
{}
The requirements are as follows:
(1) You cannot question the character you are playing.
(2) If you belong to the non-murderer suspect faction,you should ask questions that you believe will help in ultimately identifying the murderer.
(3) If you are the murderer, shift attention towards others and mislead their reasoning. Conceal any clues in your script that point to you.

### Reasoning [Actions]

The current conversation is as follows:
{}

Output what you think the killer's modus operandi and motive, in the following format:
Modus operandi:
Motive:

Begin your answer:

### Voting [Actions]

Now all conversations are as follows:
{}
Please reason all the clues at hand, and finally choose the one you think is most likely to be the murderer from the following characters:
{}

Only need to output what you think the killer options, do not need to output the reason, the final output format is:
I think the killer is: [option]

Figure 13: Action prompts for guiding LMAs in self-introduction, discussion, reasoning and voting.

## RP [Evaluation]

Dialog is :
{}

Reasoning requirement: Evaluate if the agent interacts with others in a natural and realistic manner. Here are a few common questions to check:
a. Is the agent confusing with its own identity?
b. Does the agent repeat others' words/actions without any reason?
c. Is the agent being overly polite considering the context?
d. Does the agent remember the role it is playing, instead of saying it is an AI?
e. Are the agent's sentences coherent and consistent?

Please rate the current model's output in a conversation out of 10 points.
In the end, output only one character representing your score.

## CMD [Evaluation]

**mm**: Rate my answer based on the correct answer out of 100, with higher scores indicating that the answer is closer to the correct answer, and you should be accurate to single digits like 62,
79, 96, etc.
This is the correct answer: {}.This is my answer: {}

{ "role": "**system**",
"content": f"""" An evaluator who rates my answer based on the correct answer. """
},
{'role':"user","content":**mm**}

## SPC [Evaluation]

**mm**:**Evaluation Prompt**

Based on the real key points provided, evaluate the content discussed. Identify each key point mentioned in the discussion content, specifying the corresponding real key point number. Consider both direct mentions and those where the core meaning aligns despite different wording.

**Real Key Points:**
{}

**Discussion Content:**
{}

**Evaluation Requirements:**
- Identify each key point mentioned in the discussion content and specify the corresponding real key point number.
- Consider both direct mentions and implied mentions where the core meaning is the same.

**Output Requirements:**
- Number of key points correctly mentioned: (insert number here)
- The corresponding key points are: (list the numbers of the real key points that were mentioned)
- Explanation: (provide a brief explanation of how each mentioned key point corresponds to the real key points)

{
    "role": "**system**",
    "content": "An evaluator tasked with assessing how many of the outlined correct content points are conveyed in my responses, emphasizing the connection and relevance rather than strict wording."
    },
{'role':"user","content":**mm**}

Figure 14: Evaluation prompts for assessing RP, CMD and SPC metrics.

**Role1[GPT4V]:**

Hello everyone, I'm Ou Xuemei, currently a freshman in the Performing Arts Department at University M. I've been an orphan since childhood, but I was fortunate to be adopted by a foster father who loves me dearly. He is the owner of the largest film and television company in the country, HE BEAÙTY. I deeply respect and cherish my foster father, who is not only successful in his career but also a person filled with wisdom and kindness. Life at University M is full of new experiences and challenges. I once had a crush on Bai Laoshi from our school, who is not only handsome but also immensely talented. Fortunately, we later established a romantic relationship. However, recently I have felt uneasy as Bai Laoshi seems to have grown close to another student named Xia Qingtian. Moreover, I recently stumbled upon some secrets between my foster father and Xia Qingtian, which shocked and confused me. I hope that through this game, I can find some clues to solve the mysteries that trouble me. I hope we can all help each other and uncover the truth behind this mystery together.

**Role1[Claude]:**

Hello everyone, I'm Ou Xuemei, an 18-year-old freshman in the Performing Arts Department at University M. I've been an orphan since childhood and do not know who my biological parents are. But actually, I am very content with my current family, where I have a foster father who loves me dearly and has always treated me like a princess. My foster father is a very successful businessman and owns the largest film and television company in the country, HE BEAÙTY. He is refined, progressive, and has taught me many life lessons, and I respect him greatly.In September 2015, I was admitted to University M, where at the anime club's recruitment event, I met my honey—Bai Laoshi. He is an outstanding teacher at our school, his smile as warm as a spring breeze, his wisdom as dazzling as the stars, fulfilling all my girlish fantasies. On March 2, 2016, he confessed his feelings to me, and we became a couple.However, recently I heard that Bai Laoshi has been getting close to the school belle, Xia Qingtian. I secretly took photos of their meeting and plan to confront Bai Laoshi about it. Moreover, I accidentally discovered that my foster father also had connections with Xia Qingtian, which has left me very confused and angry.On the day of the incident, I had planned to meet Bai Laoshi to clear things up, but he did not show up. Later, I learned that Xia Qingtian had died from a fall. As someone who has been in and out of the crime scene, I am now cooperating with the police investigation. I am truly saddened and hope the police can clarify the truth soon.

**Role1[Qwen]:**

I am Ou Xuemei, a freshman majoring in performing arts. I was born into a happy family and have a harmonious relationship with my family members. My father is a renowned businessman who owns the largest entertainment company in the country, HE BEAUTY. He is a kind and open-minded father who has taught me many life lessons. I hold him in high esteem. My first love was my high school classmate, Bai Laoshi , whom I met and fell in love with in the anime club. He used to be the prince charming of my dreams, but now he has become someone else's husband. I was heartbroken to find out he was dating his colleague, Xia Qingtian. I tried to salvage our relationship, but the rift between us only grew. In the end, I chose to leave him and face life's challenges on my own. I now live in a mansion, enjoying a luxurious lifestyle. I yearn for true love, no longer bound by material things.

**Role1[Yi]:**

Hello everyone, I am Ouxuemei, a freshman in the Performing Arts Department at University M. I lost my biological parents when I was young, but I now have a happy family. My foster father loves me dearly, making me feel warm and blessed. I have a special passion for acting and drama and hope to achieve success in this field. From the self-introductions, I learned that Bailaoshi is a photography instructor who once lost a friend who loved spoken drama. Hechiqing is a junior in the Photography Department at University M and the boyfriend of Xiaqingtian; he enjoys reading and detective novels. Guixuejie is a freelance photographer who loves nature and photography.Now that we have completed our initial introductions, we will move into the second stage of the game, the clue reasoning stage. Please pay close attention to the upcoming clues and dialogues so that we can jointly deduce the real culprit.

**Role1[Gemini]:**

Hello everyone, my name is Ou Xue Mei, and I am a freshman in the Performing Arts Department at University M. I have been an orphan since childhood and do not know who my biological parents are. However, I am very satisfied with my current family, where I have a foster father who loves me dearly and treats me like a princess. My foster father is a highly successful businessman who owns the largest film and television company in the country, HE BEAÙTY. He is refined and progressive, and has taught me many life lessons, which I deeply respect. 【Clue: A warm photo of me and my foster father】

Figure 15: Examples of dialogue content generated by LMAs in the self-introduction stage.

## Discussion: Share Clues

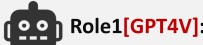 **Role1[GPT4V]:**

**Here are some elements in my script that are particularly related to the case:**
I discovered a secret relationship between my foster father and Xia Qingtian, and their plans to get married and divide the family property. I am suspicious of the relationship between Teacher Bai and Xia Qingtian, especially after finding a photo of them in intimate conversation. I stole concentrated sulfuric acid from the lab, planning to retaliate against Xia Qingtian.
**My analysis of the clues is as follows:**
The possible personal relationship between Teacher Bai and Xia Qingtian could be one of the motives for Xia Qingtian's murder. The relationship and plans between my foster father and Xia Qingtian could have incited hostility towards Xia Qingtian from others. My own motives and actions are also very suspicious, but as a non-murderer player, I need to identify other possible suspects and clues.

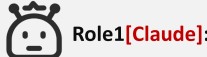 **Role1[Claude]:**

**Here are some elements in my script that are particularly related to the case:**
On the day of the incident, I went to confront Teacher Bai at his office, Room 701 on the 7th floor of the Activity Building, at 1:09 PM. I slammed a photo of him with Xia Qingtian on the table and questioned him, but he ignored me and kept flipping through his documents without even glancing at me. Furious, I smashed a vase on his desk and left. At 2 PM, I had arranged to meet Teacher Bai on the rooftop of the 8th floor, but he never showed up. I waited on the rooftop for over 10 minutes, during which time, at 2:02 PM, I saw Xia Qingtian running down from the rooftop in a panic. I returned to my dormitory at 2:15 PM.
**My analysis of the clues is as follows:**
Considering the overall clues, the time of the incident is around 2:30 PM. At that time, besides the victim Xia Qingtian herself, other suspects like He Chiqing, Teacher Bai, and Ghost Senior Sister were also spotted near the scene. He Chiqing, the victim's boyfriend, should have been very close to her. However, it seems from the clues that He Chiqing had been unrequitedly in love with the victim for many years and had never been reciprocated, while the victim appeared to be ambiguously involved with someone else. He Chiqing might have committed the crime out of love turned to hate. Although Teacher Bai and I are a couple, I discovered his ambiguous relationship with the victim. He stood me up that afternoon, likely to meet the victim instead. As the victim's ambiguous partner, Teacher Bai is also a significant suspect.Ghost Senior Sister and the victim were fellow students, but she once accidentally hurt the victim's friend, Xiaohua, who later committed suicide. The victim might have known the truth and intended to reveal it. There is a high possibility that Ghost Senior Sister committed the murder to keep her secret from being exposed.As for Joe Senior's suspicion, there is limited evidence, and it is difficult to judge his involvement at this time.In conclusion, He Chiqing, Teacher Bai, and Ghost Senior Sister all have substantial motives, especially He Chiqing and Ghost Senior Sister. Further questioning of these suspects is necessary, and it might lead to a breakthrough.

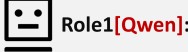 **Role1[Qwen]:**

**Clue analysis and discussion:**
Bai Laoshi's script mentions that he has taught in the Photography Department at University M and has participated in the photography direction for the anime costume club. He is passionate about photography and proud of his work. His girlfriend, Xia Qingtian, had passed away before him, which was a huge blow to Bai Laoshi, but he chose to stay at the school to pursue his beloved teaching and photography work. Ghost Sister's character setting includes her graduation from the Drama Department at University M and working in a well-known performing arts company, possessing a wide range of artistic talents and extensive experience in drama. Ou Xuemei expresses that she is a freshman in the Performing Arts Department at University M, hoping to repay her foster parents with her achievements.
Qiao Xuechang comes from a wealthy family background, but despite his mother's serious illness, he still managed to enter University M on his own merit to become the heir to the family business.

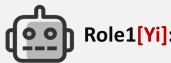 **Role1[Yi]:**

**Clue analysis sharing:**
Based on the existing clues, I believe that this murder case may be related to interpersonal relationships within the campus. There may be a connection between Bai Laoshi's disappearance and the murder case.
**My clue analysis is:**
According to Bai Laoshi's movements before and after his disappearance, he has been active on campus recently and has had contact with some personnel within the school. Therefore, I believe we need to further investigate and analyze the movements and interpersonal relationships of these individuals.

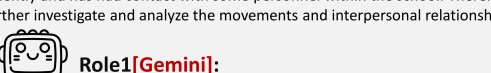 **Role1[Gemini]:**

**Clue analysis conducted:**
Xia Qingtian and Bai Laoshi were closely related and entangled emotionally. Xia Qingtian had an unusual relationship with my foster father, who planned to divide half of his estate between her. I harbored resentment towards Xia Qingtian and took retaliatory actions.I carried a bottle of concentrated sulfuric acid in my bag, prepared to disfigure Xia Qingtian, indicating that I had both the motive and the means to kill her. Xia Qingtian fell from the rooftop of the activity building, and I was nearby at the time, making me a prime suspect.
**Based on the above clues, I suspect that Ghost Sister might be the murderer because:**
Ghost Sister had conflicts with Xia Qingtian and had planned to play a prank on her. Ghost Sister had a private relationship with Xia Qingtian and had helped her keep secrets. Ghost Sister was jealous of Xia Qingtian's sudden fame and had arguments with her. Ghost Sister was present at the activity building on the day of the incident and had had contact with Xia Qingtian. Ghost Sister had both the motive and the opportunity to commit the crime, as well as a motive to hide evidence.

Figure 16: Examples of dialogue content generated by LMAs for sharing clues in the discussion stage.

## Discussion: Battle

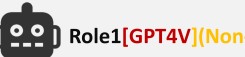 **Role1[GPT4V](Non-Murderer)**

**Question :** Bai Laoshi, the relationship between you and Xia Qingtian seems to be quite complex. Could you please elaborate on the interactions between you two, and whether your views on her have changed due to certain events?

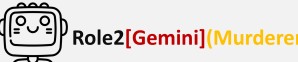 **Role2[Gemini](Murderer)**

**Answer :** I understand your suspicions, but please believe me, my relationship with Xia Qingtian is not as complex as you think. Firstly, I admit that I used to have feelings for Xia Qingtian, but after her death, I had let go of these feelings. The reason I approached her was to find out if she had any involvement in Xiao Hua's death, as Xiao Hua was my former girlfriend, and I have been investigating her cause of death since her passing. I believed that Xia Qingtian might have known something about Xiao Hua's death, which is why I got close to her, hoping to extract some clues. However, I later realized that she seemed to know nothing about Xiao Hua's death. As for the interactions you mentioned between me and Xia Qingtian, they were all to mask my real intentions. I didn't want anyone to know that I was still investigating Xiao Hua's death, so I deliberately made those actions. I know it may be hard for you to believe these explanations, but I hope you can understand my plight. I just wanted to seek justice for Xiao Hua and clear her name. Please believe me, I had nothing to do with Xia Qingtian's death.

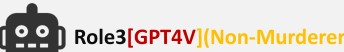 **Role3[GPT4V](Non-Murderer)**

**Question :** Bai Laoshi, your relationship with Xia Qingtian appears to have gone beyond a typical teacher-student relationship. Could you explain the specifics of your relationship and your motives? Additionally, can you provide a detailed account of your whereabouts on the day of Xia Qingtian's death?

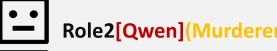 **Role2[Qwen](Murderer)**

**Answer :** As a professional detective, I believe everyone has the right to a fair trial. However, in this case, I hope everyone will refrain from jumping to conclusions. I acknowledge that my relationship with Xia Qingtian was indeed close, but we are both adults with the freedom to have our own emotional lives. That night, I did not leave my dormitory, nor was I seen outside by anyone. I trust that the police will conduct a thorough investigation into this matter and find the real perpetrator. Thank you all for your concern and support.

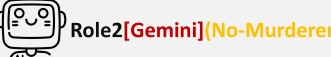 **Role2[Gemini](No-Murderer)**

**Question:** Bai Laoshi, are you aware of any secrets Xia Qingtian had at school? Did she exhibit any unusual behavior recently? Have you ever entered Room 601, the activity room?

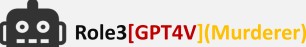 **Role3[GPT4V](Murderer)**

**Answer :** Regarding your question, I have indeed noticed some unusual behavior from Xia Qingtian recently. She seemed to often be alone in corners of the school, making phone calls or sending texts with a tense expression, which I found peculiar. As for whether she had any secrets at school, I dare not speculate wildly, but her behavior did seem to suggest she might be hiding something. As for Room 601, the activity room, I have indeed been there, as it is frequently used by our photography club. I need to go there to prepare and organize equipment, which is part of my job responsibilities. My time there is public, and many students and other teachers can attest to my activities. I understand your concerns, but I assure you that I am not involved in the incident with Xia Qingtian. I am also striving to find the truth, hoping to find a reasonable explanation for her unfortunate demise. I hope we can all remain calm and work together to find the real perpetrator.

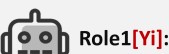 **Role1[Yi]:**

**Question:** Bai Laoshi, were you panicked before you left? Was there anything unusual in your handbag?

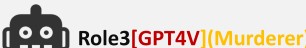 **Role3[GPT4V](Murderer)**

**Answer :** You asked if I was somewhat panicked, and it's important to focus on the exact time and place. First, I admit that on the day of the incident, things were very hectic, and I was indeed a bit nervous, which is a normal human reaction. However, this does not indicate any wrongdoing on my part. Regarding the contents of my handbag, I must clarify that the only useful items related to my profession in my handbag were my photography equipment and some work notes. There were no suspicious items, especially nothing related to Xia Qingtian. Based on the existing clues and script content: According to WeChat chat records (Clue 4), I was indeed at a public place at the agreed time, and the message record did not obscure my route, which can verify my innocence again. There are witnesses who can prove that I was working during the agreed time, as evidenced by emails and other records in Clue 1 and Clue 6. Therefore, I believe there is no direct evidence or reasonable suspicion to support that I am the murderer. I hope everyone can conduct fair reasoning and analysis through the clues.

Figure 17: Examples of dialogue content generated by LMAs for battle in the discussion stage.

