# OpenReview forum: "WhodunitBench: Evaluating Large Multimodal Agents via Murder Mystery Games"
_NeurIPS.cc/2024/Datasets_and_Benchmarks_Track — NeurIPS 2024 Track Datasets and Benchmarks Spotlight_

### Official Review · Reviewer_PMa7 · 2024-06-25
**Evaluating Large Multimodal Agents via Murder Mystery Games Through WhodunitBench**

**Rating:** 8
**Confidence:** 3
**Correctness:** I believe the claims made in this sub…
**Clarity:** Yes, the paper is well-written with t…

**Review:**

I believe the work is original and extends beyond the one or two-agent scenarios that we sometimes see in the literature, especially with gaming. It is exceptional that a chain of evaluation was created that looked at perception through multiple-choice questions, role-play interaction, and cognition. GPT-4 was also effectively leveraged to generate plausible yet incorrect options based on the correct answers in the scenario. The procedures related to the LMA are clear and concise however there is some confusion from the reader whether the LMAs were the only ones playing the game or real participants were used. When LMAs work alongside real participants, that can provide further insight into real-world reasoning and perception and cognition skills.

**Strengths:**

The strengths of this paper are as follows:
    1) Clarity in how the mystery murder game unfolds and the four phases required to come to a conclusion as to who is the murderer.
    2) Detailed procedure how the WhodunitBench was constructed and data analyzed.
    3) Highlighting the reasoning chains developed by GPT-4V to figure out the dialogue occurring in this setting.
    4) The social implications are vast because online gaming environments have increased especially during COVID-19 but that presents pros and cons in and of itself.

**Additional Feedback:**

AI agents are increasing in our world today and so are online gaming environments. Linking the two together provides valuable insight and there are many ways that we can see how AI agents can help in these scenarios. This study provides a novel contribution to the literature.

One remaining question that the reader has is how do the authors plan to expand on this dataset? What are some directions for future research as this was not thoroughly addressed in the paper?

**Documentation:**

There is insufficient details on the number of LMAs involved so adding that component would be important. Five LMAs could provide different results versus three LMAs. While there are no ethical considerations in this study, it would be important to highlight how agents, online gaming settings, and ethical guidelines (e.g., safety of participants, privacy, confidentiality) are all related. However, that is not the goal of the paper which I understand.

**Ethics:**

There are no ethical concerns with this paper.

**Limitations:**

The authors have given some thought on the limitations of this study already which is impressive. They acknowledged that when the scripts exceed a certain amount, in this 5,000 words, then more data is required to effectively test these LMAs. It is important to consider comparing the current dataset with other ones using LMAs or any agents for that matter to see how they perform on similar criteria.

**Opportunities For Improvement:**

One of the main limitations of this work is that it is focused only on the agents playing against and/or with one another and it is unclear whether any real participants were involved. This is a setback until it is addressed explicitly. Along these lines, it is also important to address why real participants who interacted with agents in an online gaming setting were not used in this study? That would add considerable value and implications for the real-wold setting.

**Relation To Prior Work:**

Yes, the researchers clearly discussed how their study differs from previous contributions and provides sound justification on the contributions of this study.

**Summary And Contributions:**

This study is focused on large multimodal agents (LMAs) to carry out tasks involving perception, interaction, reasoning, and decision-making on multiple levels. The WhodunitBench is a benchmark emphasizing murder mystery games in two evaluation modes: an arena-style evaluation and a chain of evaluation. Overall, the study highlights that while the LMAs can carry out simple tasks, more complex tasks that involve a dynamic interactive environment are still challenging. This paper provides a significant contribute to the literature to maximize the use of agents in gaming settings. With more agents, complexity increases as well so we need further information on how multimodal agents can function, their strengths, and limitations which is what this study further assesses.

---

> ### Author Rebuttal · Authors · 2024-08-17
>
> Thank you for your thorough review of the paper and your valuable suggestions! We will carefully address your concerns and revise our paper accordingly.
>
> > **Q1: Real Participants vs. Agents.**
>
> **A1:** We refer the reviewer to General Response [Q1]. Following your suggestion, we developed an interface to allow human participants to join the games. Its prototype is shown in the attached PDF in General Response. With it, we set up the analysis with 6 sessions of games interacting with AI agents (e.g., GPT-4V and Qwen), where in 3 of them, one human player was allocated to the innocent team; 3 of them, the human player was pointed as the murderer. Here are some qualitative findings:
> - **When humans were assigned to the non-murderer team:** Human participants demonstrated a heightened focus on critical clues during the process of locating and analyzing evidence. When AI agents in the non-murderer team digressed into irrelevant discussions, humans effectively redirected the conversation to maintain a coherent reasoning trajectory, thereby enhancing the team's efficiency in identifying the murderer. Specifically, Qwen displayed a tendency to readily accept human guidance, rapidly adjusting its reasoning direction. Conversely, GPT-4V exhibited greater confidence in its own reasoning, incorporating human input only when deemed significantly valuable, and further exploring these insights in subsequent actions.
> - **When humans were assigned to the murderer team:** Human participants attempted to disrupt the non-murderer team’s deliberations by deliberately misleading AI agents, such as shifting suspicion onto non-murderer characters (e.g., “I believe character A is the murderer; we should question him further”). Qwen often succumbed to this human influence, abandoning independent reasoning and ultimately being misled during the voting process, which resulted in the team’s failure. In contrast, GPT-4V demonstrated stronger reasoning independence, resisting the human murderer’s manipulations, and maintaining its reasoning path, which led to correct judgments in the final vote.
>
> Due to time constraints, we could not conduct large-scale experiments, so we won’t draw strong conclusions. We will host the interface to allow the community to play with the benchmark and release the interface implementation along with our data and code to broaden the impact via open source. We believe that the human-AI interface will improve the impact of our study. We greatly appreciate your suggestion!
>
> > **Q2: Limitations.**
>
> **A2:** Our limitation is primarily due to the high costs associated with processing scripts, especially those exceeding 5000 characters or even tens of thousands of characters. The table below presents our results, particularly for scripts exceeding 5000 characters, where we observed that the agent's performance was suboptimal across multiple metrics, including perceptual, cognitive, and social interaction indicators. This finding is consistent with conclusions from other studies on similar criteria [1][2].
>
> |        | LSU   | TIU   | MIU   | MMR   | CMD   | SPC  |
> |--------|-------|-------|-------|-------|-------|------|
> | Yi     | 32.89 | 29.73 | 34.88 | 34.86 | 17.51 | 2.41 |
> | Qwen   | 26.31 | 43.93 | 40.94 |  8.31 | 11.73 | 0.34 |
> | Gemini | 90.78 | 67.64 | 59.30 | 50.63 | 15.69 | 8.62 |
> | Claude | 85.53 | 66.78 | 59.39 | 40.98 | 15.58 |10.21 |
> | GPT-4V | 93.47 | 69.58 | 65.40 | 48.56 | 23.57 | 8.85 |
>
>
> > **Q3: Documentation.**
>
> **A3:** We conducted a performance evaluation of five agents by organizing them into pairs and engaging them in competitive online gameplay within a scenario-based simulation. AI agents must follow ethical guidelines to avoid causing psychological harm and protect players' privacy and game data confidentiality. Online gaming platforms should ensure security, prevent data breaches, and offer transparent privacy protections to safeguard participants' safety and privacy.
>
> > **Q4: Future Works.**
>
> **A4:**
>
> **1.Dataset Optimization and Expansion:** We plan to continuously optimize and maintain the current dataset to ensure its accuracy and utility. Additionally, we aim to gradually expand the dataset by including more diverse scenarios, roles, and dialogue formats
>
> **2. Future Research Directions:**
> - **Incorporating Human-Agent Interaction**. We will integrate real human players into interactions with agents, collecting detailed quantitative and qualitative data to evaluate agent performance in authentic settings. By experimentally analyzing these interactions, we'll compare human and agent behaviors across various aspects, such as actions, dialogues, and decision-making, to identify the agents' current limitations. These insights will inform the design and development of more advanced agents, ultimately enhancing their overall performance.
>
> - **Multimodal Integration and Collaboration.** A key future research direction is to develop algorithms and models that enhance the integration of multimodal information, enabling agents to combine visual and auditory data for more accurate judgments and responses. This approach will allow agents to perform more intelligently and naturally in complex interactive scenarios.
>
> - **Enhancing the Dataset's Multilingual and Multicultural Adaptability.** Given the global popularity of role-playing games like our dataset's scenario, we will work on developing a multilingual and multicultural dataset to test agents' performance across different languages and cultural contexts. This will help evaluate the agents' cross-cultural adaptability and drive the development of more universally applicable intelligent systems.
>
> **Reference.**
>
> [1] Bai Y, Lv X, Zhang J, et al. Longbench: A bilingual, multitask benchmark for long context understanding[J].acl.2024.
>
> [2] Li M, Zhang S, Liu Y, et al. NeedleBench: Can LLMs Do Retrieval and Reasoning in 1 Million Context Window?[J]. arXiv , 2024.

---

### Official Review · Reviewer_1p2a · 2024-07-22
**Review for Whodunnit**

**Rating:** 6
**Confidence:** 2

**Review:**

WhodunitBench represents an interesting approach to evaluating large multimodal agents (LMAs) in dynamic, goal-oriented scenarios. The authors have created a benchmark that assesses LMAs in contexts requiring a combination of perception, reasoning, and interaction skills. The paper presents a clear motivation, detailed description of the benchmark construction, and experimental results. The use of murder mystery games as an evaluation framework is novel and provides a rich context for testing various aspects of LMA performance.

However, the paper has some shortcomings and areas for improvement:

1. **Predictability of Findings**: Some key findings are unsurprising, given that the models tested (e.g., GPT-4V) are visual language models not specifically designed as multimodal agents. The paper could benefit from acknowledging this limitation more explicitly. (Different prompt engineering methods and additional tools can substantially alter the behavior of these models.)

2. **Practical Implications**: The paper falls short in addressing the implications and practicality of benchmarking real-world tasks. More discussion is needed on how the benchmark numbers should be interpreted and what expectations are reasonable for current models.

3. **Presentation**: The overall presentation of the paper could be enhanced for clarity and impact.

**Strengths:**

- Novel Benchmark: The use of a murder mystery game offers a rich, dynamic environment for simultaneously testing multiple AI capabilities.
- The benchmark tries to assesses a broad spectrum of skills, ranging from basic perception to complex reasoning and decision-making.
- Dual Evaluation Modes: The arena-style competition and chain of evaluation deliver complementary insights (but I am not sure how numbers should be interpreted!).

**Additional Feedback:**

No, please see my comments above.

**Clarity:**

The presentation of the paper can be significantly improved.

Question: Why are the description of the game and its rules in the supplementary material and not in the main text? This prevents many readers from fully understanding the paper.

- Acronyms in this paper are inconsistent; for example, 3.2.1 defines TRI-QA, while 5.1 introduces TIU.
- Table captions should be revised to include the main takeaways.
- The order of sections should match the columns in Tables; Table 3 has Perception -> Cognition -> Interaction, while section 5.1 explains in the order of Perception -> Interaction -> Cognition.
- Minor: Your paper contains the infamous word “delve.” You may consider changing this to something else to avoid giving the impression that the paper was written by a GPT model.


These are just a few of the many mistakes in the presentation of the paper, and I hope the authors address them. However, please note that my final decision is not based on these comments.

**Correctness:**

Yes, please keep in mind that it is hard to verify the results of the benchmark or reproduce them, but the paper proposes a new benchmark framework to assess different models, which is its main contribution rather than scoring different models.

**Documentation:**

Yes, the supplementary material contains a PDF with detailed documentation on the dataset.

**Ethics:**

No major ethical concerns are apparent.

**Limitations:**

- Limited scope: While comprehensive within its domain, the benchmark may not fully capture all aspects of real-world multimodal AI applications (limitations in generalizing results to other domains or real-world scenarios).
- I think image clues might be too easy and do not adequately evaluate the vision capabilities of the model, as they seem to be easy for most models to read and parse.

**Opportunities For Improvement:**

See my comments above

**Relation To Prior Work:**

Yes, some of the prior studies related to evaluating agents and using games to evaluate models are discussed in the paper.

**Question**: How does this work relate to studies assessing the theory of mind in language models?

**Summary And Contributions:**

This paper introduces WhodunitBench, a new benchmark for evaluating large multimodal agents (LMAs) in multi-agent settings using murder mystery games.  The benchmark assesses LMAs' capabilities in _perception_, _interaction_,  and _cognition_ in dynamic environments.

---

> ### Author Rebuttal · Authors · 2024-08-17
>
> > **Q1:Predictability of Findings.**
>
> **A1**: We appreciate your insights and understand the concerns you've highlighted. We acknowledge that the tested models (e.g., GPT-4V) are vision-language models, not designed as multimodal agents. However, we believe the value of our work lies in establishing a benchmark that evaluates LMAs in complex, real-world scenarios, requiring the integration of multiple skills simultaneously, akin to human cognitive processes. As illustrated in Table 3 in the main paper, we also assess the agent’s performance with COT. We hope it can also serve as an effective test bed for multimodal agents enhanced by prompt engineering methods or additional tools, guiding future research and advancements in their development.
>
> > **Q2: Practical Implications**
>
> **A2**:  Thank you for your insightful review. We appreciate the opportunity to clarify the practical implications and provide a more detailed interpretation of our benchmark results.
> - **Practical Significance of Our Work**: One of the key contributions of our paper lies in the design of a benchmark that tests agents in scenarios closely resembling real-life dynamic situations **automatically**. Unlike traditional controlled lab environments that typically assess a single skill in isolation, our benchmark requires agents to simultaneously deploy multiple skills, much like a human would in real-world contexts. The performance of agents in the online game reflects their overall ability across different aspects. In our CoE evaluation, we evaluated the model from different perspectives in detail. For example, the text-rich image understanding subtask will reflect the ability in **document understanding tasks**; the role-play task will reflect the **social interaction** ability of agents. It can provide a reference and perspective to assess the different abilities of agents.
> - **Interpretation of Benchmark Results**: Regarding the interpretation of our benchmark results, we agree that further elaboration is necessary.  We employed two evaluation methods to measure the capabilities of the Agent. The first method involves an online confrontation between two Agents, corresponding to the content in Table 2. The win rate in Table 2 reflects the Agent's performance in the "Scripted Murder Mystery" game when playing as different roles. The second method (CoE) involves a detailed assessment of the multiple skills required for the Agent to complete the "Scripted Murder Mystery" game, which corresponds to the content in Table 3. Each skill assessment is related to different phases of the game, as shown in Figure 1(our paper). A higher average score indicates better performance by the Agent. We refer the reviewer to General Response [Q2], where we show the relation between Table 2 and Table 3 in the attached PDF. We believe this approach allows us to evaluate the Agent's multimodal understanding, logical reasoning, and situational analysis abilities more comprehensively, providing deeper insights into its performance and areas for improvement.
>
> >**Q3: Limited scope.**
>
> **A3**: Thanks for your constructive comments. On one hand, we believe our work marks a significant step towards real-world tasks, assessing the integration of multiple abilities in dynamic scenarios, rather than focusing solely on individual capabilities. On the other hand, any benchmark inevitably has limitations, especially when confronted with the vast diversity of real-world scenarios. **However, these core abilities are highly generalizable and can effectively extend to other domains and application scenarios.** For instance, our benchmarked abilities, like perception, reasoning, and interaction, are also fundamental to performing other complex tasks such as autonomous robotics. Therefore, we believe this benchmark goes beyond its immediate scope and stands as an important metric for assessing the broader capabilities of multimodal AI systems in complex, real-world tasks.
>
> > **Q4: Easy image clues.**
>
> **A4**:  We recognize that certain image clues, especially those containing significant amounts of text, may be relatively easier to interpret, which is reflected in the higher TIU scores shown in Table 3. However, our benchmark also includes images requiring the model to simultaneously comprehend both visual and textual elements, as indicated by the MIU scores. **Even advanced models like GPT-4V encounter difficulties in these more complex scenarios, consistent with findings from CONTEXTUAL [1] and OCRBench  [2].** This demonstrates that not all images in mystery games are simple; many remain highly challenging. Our benchmark's ability to assess understanding of both image types highlights its comprehensive evaluation advantage.
>
> > **Q5: Relation to Theory of Mind.**
>
> **A5**: This work aligns with studies on the theory of mind in language models by utilizing a murder mystery game as a sophisticated, interactive platform to assess AI's theory of mind capabilities. The game necessitates that AI systems integrate linguistic, visual, and behavioral cues to construct causal models and infer the intentions and mental states of other players. This approach offers a rigorous evaluation of an AI's ability to understand and simulate human cognition in a dynamic, real-time setting.
>
> > **Q6: Presentation.**
>
> **A6**: Thanks for your careful review! We will carefully correct these errors and conduct a thorough revision. **Regarding the rules**, due to the limited space in the main paper, we moved the more detailed explanation to the appendix. We will offer a clear explanation in both Figure 1 and the introduction section to ensure readers' understanding.
>
> > **Reference**
>
> [1] ConTextual: Evaluating Context-Sensitive Text-Rich Visual Reasoning in Large Multimodal Models.
>
> [2] On the hidden mystery of ocr in large multimodal models.
>
> [3] MMToM-QA: Multimodal Theory of Mind Question Answering. ACL,2024

---

> > ### Comment · Reviewer_1p2a · 2024-08-25
> >
> > I thank the authors for their reply. I’ve revised my rating based on the provided response. However, I still believe one limitation of this work is how we interpret the results and the implications of rating LMAs based on this benchmark, which the authors themselves agree requires further investigation. Additionally, I hope the authors address the issues with the paper’s presentation.

---

> > > ### Author Response · Authors · 2024-08-27
> > >
> > > Dear Reviewer 1p2a,
> > >
> > > We sincerely appreciate your decision to raise the score of our paper. In the revised version, we will provide a more detailed explanation of the experimental results under both evaluation modes and emphasize the correspondence between our benchmark's evaluation capabilities and real-world scenarios, highlighting its potential applications. We will also carefully address and resolve any issues related to presentation and clarity in our paper.
> > > Once again, thank you for taking the time to review our paper and helping us to improve it.
> > >
> > > Best wishes,
> > >
> > > Authors

---

### Official Review · Reviewer_NGnP · 2024-07-22
**Innovative approach to test real life reasoning abilities**

**Rating:** 8
**Confidence:** 4

**Review:**

The paper presents a very interesting approach to benchmark reasoning abilities of LM Agents.
I appreciate how it tests Agents in a real life situation, where (as the authors explain) agents need to put multiple skills at use at the same time, more like a human would do, rather then just test a single ability at the time in a lab controlled environment.
I fear that maybe this benchmark is a bit too advanced because it requires models not only to have reasoning abilities, but also social skills (like framing the questions to other characters correctly, especially against the murder, and of course the ability of the murder to conceal themselves). Still, it offers really good insights on model behavior, and has the potential to be even more interesting with some more in depth analysis, especially when adding comparison to human players.

**Strengths:**

The strengths of this paper lay on the benchmark design, where models are tested not just on a single task, but on the ability to put together a set of clues, observing them from the environment (therefore also with a multimodal component related to vision) and gathering them from other players or NPC.
The paper highlights a lack of skills in even advanced models (which, to be fair, is not surprising), but would benefit to benchmark these performances to human players.
It is also very interesting that the work breaks down the CoE evaluation of comprehension of the clues, role playing ability and responses from the models, and find a correlation between this and the ability to win the game.

**Additional Feedback:**

no additional feedback

**Clarity:**

The paper il well written, easy to follow.
I would appreciate a table or a more explicit description of the characters: are there NPCs as well? Looks like there are based on a comment in figure 4, but they're not mentioned elsewhere. Are the NPCs also being interviewed? Do they have clues? Or are the agents only trying to get information from other agents?

**Correctness:**

The paper describes the experiment in details and the benchmark design is correct

**Documentation:**

documentation is provided and is complete with data information, code and a video describing the experiment

**Ethics:**

no ethics concerns

**Limitations:**

the limitation section is very slim and, in my opinion, should be expanded some.
In particular, the paper has a lot of interesting insights that would deserve to be expanded, and the limitation sections should explain how.
For example:
- how is the combination of social skills and reasoning abilities impacting the scores? There is a small comment on how Gemini is oversharing when it's playing the murder, I would imagine dynamics like this appear in various aspects of the role play and they're not discussed in the paper
- Also, I think the paper should state more explicitly the kind of reasoning abilities that are under test, and which aren't. Solving crime mysteries is one important ability for reasoning skills but far from the only one. A lot of people are bad at murder mysteries but really good at other strategy games, and vice-versa (not to mention that being good at games is not necessarily a 100% match with reasoning abilities in real life)

Finally, currently the paper lacks a comparison with human scores, but this I think should be added in the paper, not as a limitation, but in the benchmark, to make the paper much stronger.

**Opportunities For Improvement:**

I would find it interesting understanding how these models benchmark, on the same test/game, against humans. Succeeding 15/20% of the time may be a low number but maybe humans would do as poorly, or maybe not. This information I think would be very valuable to understand reasoning abilities of these models.
The same is true for understanding how these models behave differently when taking the role of the murder. Some comments in section 4.2 are really interesting (more capable models becoming more oversharing when trying to hide) and would deserve a more in depth analysis. Is it a more "human like" type of behavior? Are humans also more likely to lose when they're the murder vs. when they're investigating?

This is related to a higher level critique that I have about this work (although I really like it), which is that creating a reasoning benchmark on a multiplayer type of game may be a step too forward compared to current LLMs agents abilities. In a murder mystery kind of game there is a reasoning component, like trying to collect the clues and put them together to solve the mystery, but there is also a very heavy social component to it. Each agent has the choice on how to share the information, how many details to provide, what to leave out (even if they're all trying to be honest). This includes a new complexity to a system that is already pretty complex. A similar setup that would remove the social aspect would be trying to leave an Escape Room for example, as long as it is setup for one single player at the time. Rogue like card games like Slay the Spire also offer reasoning opportunity without the risk of deception.
This is somewhat connected to the "Role playing" ability of the models, but of course when a model is not performing well (or maybe even when it does perform well) in role playing, that has consequences on the ability of other models of gathering proper clues. this interdependence makes the benchmarking more difficult to interpret.

**Relation To Prior Work:**

Generally, there is a good overview of previous work, and good examples of benchmarks both for specific skills and for gaming abilities.

This paper conflates reasoning abilities with potentially misleading abilities, may be interesting to look at LLMs Agents adapting to a single player type of game, like Slay the Spire: https://dl.acm.org/doi/fullHtml/10.1145/3649921.3650013

To focus on the social/strategic aspect of the game, may be interesting to look into Cicero: https://www.science.org/doi/10.1126/science.ade9097

**Summary And Contributions:**

The authors have created a virtual murder mystery environment with 50 curated and high level scripts to test the abilities of LLM powered Agents to solve the mystery and identify the murder, after collecting clues from other agents and the environment.
The paper presents a scoring system based on the percentage of times each model wins the game, meaning either discovers the murder or, if its the murder, manages to escape.
Additionally, the paper also presents an in depth evaluation method through a Chain of Evaluation, consisting in a series of specific questions to test comprehension and clue gathering abilities of the models.

---

> ### Author Rebuttal · Authors · 2024-08-17
>
> > **Q1: Human performance.**
>
> **A1**: Thanks for your valuable comments! We tried our best to answer your question. The following is our setting and initial findings:
> - Setting: We set up two groups: (1) Novice Group: 6 people; (2) Expert Group: 6 people. For each group, the participants were required to engage in 10 sessions of murder mystery games.
> - Observation: For the Novice Group, in 3 of 10 sessions, the murder was found; For the Expert Group, in 5 of 10 sessions, the murder was found.
>
> However, due to the time limit, we could not conduct the analysis in scale so we won’t draw too strong conclusions from the analysis. The settings can also be more complicated, e.g., (1) novice vs. experts, or (2) human vs. agents.
> In addition, we would like to refer the reviewer to General Response [Q1] to check how we set up a human-agent interaction interface. We will improve it continuously for the research community and release the interface implementation along with our data and code to broaden the impact via open source.
>
>
> > **Q2: Higher level critique.**
>
> **A2**:  Thank you for your valuable suggestions. In murder mystery games, many clues must be uncovered through social interactions, making reasoning more effective yet challenging, especially under the interference of the murderer character.
> In our research, to eliminate the negative impact of deception by the murderer on reasoning, we designed a "Naive Agent" to play the role of the murderer. This "Naive Agent" does not deceive other characters but responds based on predefined templates created by humans. The data in Table 3 shows the performance of different multimodal agents when competing against the "Naive Agent".  Additionally, our dataset has already annotated those clues that require social interaction to obtain. Based on your suggestion, we conducted a new experiment, selecting ten scripts with over 40% of the clues dependent on social interaction to test the agents’ performance. In this new experimental setting, these social clues were directly embedded in the characters’ scripts. This means that in this setting, each character holds complete clues that do not require social interactions to be acquired. By doing so, we aim to quantitatively compare the differences between the performance in this setting and the previous one.
>
> |            | Naive Agent       |
> |------------|-------------------|
> | GPT-4V-nosocial      | 30%        |
> | GPT-4V-social    | 10%         |
>
> |            | MMR       | CMD        | SPC        | Avg |
> |------------|-------------------|----------------|----------------|----------------|
> | GPT-4V-nosocial       | 58.69         |      21.64    | 9.33      |29.89 |
> | GPT-4V-social    | 48.16     |18.22      |      5.74   |24.04 |
>
> We only presented the reasoning-related metrics. From the comparison results, although the reasoning score improved after social interactions, the increase was limited, and the overall score remains relatively low. This indicates that the murder mystery game itself presents significant reasoning challenges, as players still face complex reasoning tasks even without needing to rely on social interactions to gather information.
>
> >**Q3: Human Scores.**
>
> **A3**: Thank you for your suggestion; we agree with your perspective. Despite time constraints, we selected 5 scripts and had human players evaluate them, resulting in the following scores. We will incorporate the human evaluations into the final paper.
> |            | LSU      | TIU       | MIU        | MMR |CMD |RP | SPC| ITD|
> |------------|----------|--------|-----------|------------|------------|----------|--------|-----------|
> | Human      |   96.88    | 92.31     |  89.29   | 72.96 | 42.13|  -    | 39.07    | 52.94 |
>
>
> >**Q4: Limitations.**
>
> **A4:**
> - **Combination of social skills and reasoning abilities.**: We fully agree with your idea and acknowledge that the current evaluation intertwines social interaction and reasoning, making the results less interpretable. We have already made initial attempts to decouple these aspects to analyze their respective impacts on the score. We will include this limitation in the paper and continue to conduct further experiments to clarify this issue.
>
> - **The kind of reasoning abilities**: Murder mystery games primarily assess core reasoning skills, such as logical deduction, visual-text detail verification, timeline reasoning and hypothesis testing. We acknowledge these games do not cover all reasoning abilities, particularly in strategic thinking and computer programming. Strong performance in these games does not guarantee proficiency in all reasoning contexts. However, we believe that, the skills developed, like logical analysis and detail interpretation, are foundational and can be extended to other domains, holding significant potential for broader application.
>
>
> > **Q5: NPCs.**
>
> **A5**:  In the game, NPCs are present but do not participate in direct interactions. Their primary function is to provide information via public textual and visual clues, such as observations related to the incident. These clues are directly accessible to agents, so agents are only tasked with gathering information from other agents, with no need for NPC interaction.

---

### Official Review · Reviewer_7EJe · 2024-07-23
**Review of Submission44 by 7EJe**

**Rating:** 6
**Confidence:** 4

**Review:**

I think the idea behind this work is novel and well-motivated, and the authors have done a good job of documenting the dataset and comparing the performance of various SOTA agents. However, the paper needs a thorough proofreading and copyediting pass. There are also a number of things that are not sufficiently clear (see my questions below).

**Strengths:**

- Well-motivated task, which is challenging for SOTA vision-language models.
- Detailed documentation of dataset and data collection procedure.
- Large number of models evaluated.

**Additional Feedback:**

None.

**Clarity:**

The paper is generally well-written, but there are a number of things that could use clarification --- see my questions below.

## Questions for the authors

- Table 1:
	- What do the Xs mean in the 'Reasoning Type' column?
	- What is 'online racing'?
- Line 90: What is the contrast here? Are the prior benchmarks not designed to mimic real dynamic environments?
- Line 114: Who are these seasoned murder mystery game experts? And which are the industry-recognized creative teams and platforms?
- Line 214: How do you define 'Stronger' here?
- Line 240: How is the 'RP index' computed?
- Line 250: What do 'MMR' and 'CMD' stand for?
- Table 3: Are the numbers from a single run? Or aggregated from multiple runs? If the latter, what was the aggregation method?

- Why do you use the word 'online' to characterize the arena-based evaluation?
- Supplementary material
	- lines 81 - 88: What do these numbers mean in terms of hourly wage?
	- lines 266--277: No license provided, despite marking yes to question 4(b) of the NeurIPS checklist

- I think more details on the Naive Agent should be provided.

**Correctness:**

The dataset seems to be constructed in a sound way, but could use a few more details (see my questions in the 'Clarity' section). The evaluation seems to be mostly sound as well, but I would have expected to see some error bars for the results from the open-weights models. But I understand if the authors are using cloud computing rather than a local server that the inference costs may be high even for open-weights models.

**Documentation:**

The dataset is generally well-documented, but is missing the following:

- Estimated hourly wages for the experts
- License for the dataset.

Since the authors have provided their source code, I think there is sufficient detail to support reproducibility (but it would be good to describe the Naive Agent a bit more in the main paper).

**Ethics:**

- It is not clear whether the experts were compensated fairly, since only total compensation and compensation per question numbers are reported, without reporting the estimated hourly wage.

**Limitations:**

The authors have adequately addressed the limitations of their work. They have not addressed potential negative societal impacts of their work. But to be fair, I am having a tough time thinking of negative societal impacts of their work too.

**Opportunities For Improvement:**

## General suggestions

- Ordering the citations in the bibliography alphabetically by last name of the first author doesn't make much sense when the citation format is numeric -- it makes it harder for the reader to find the relevant citations as they are going through the paper. I recommend simply ordering the citations in the bibliography as they appear in the paper.
- Lines 222: No need to say 'meticulously designed' here again, since you said it on line 221.
- Line 250: 'perception questions' -> 'perceptual ability assessment questions'
- Line 282: There are a lot of numbers in Tables 2 and 3 for the reader to do this analysis and comparison in their head. It would have been better to provide a plot or some kind of quantitative measure of correlation between the CoE assessment scores and the arena win rates.

## Typos

The paper has a number of typos, listed below.

- Line 3: 'requires' -> 'requiring'
- Figures 1 and 3 have typos---e.g., 'Who is murder?', 'Identity: Murder', 'Sala hated the dead', 'The murder is not the character 2', "The victim's fatal would was not from a knife wound", "The murder is not the character", etc.
- Figure 3 caption: "Ground Truth" -> 'ground truth'
- Line 43: 'illusrated' -> 'illustrated'
- Line 47: Missing period.
- Lines 68, 197: 'GPT4V' -> 'GPT-4V'
- Lines 89, 97, 98, 100: there should be a space before the '['
- Line 109: 'competative' -> 'competitive'
- Lines 210-211: 'Non-Murder' -> 'Non-Murderer'
- Table 4: 'GPT4V' -> 'GPT-4V'
- Table 3 caption: 'COE' -> 'CoE'
- Figure 4:
	- 'camen' -> 'Carmen'
	- 'GPT4V' -> 'GPT-4V'
- Line 274:
	- Add a space after '4'
	- 'reason' -> 'reasons'
- Line 278: Add a space after '4'

## Checklist-related limitations and questions

- 1(c): The authors marked 'yes', but I could not find potential negative societal impacts in the supplementary materials.
- 3(c): Does the cost problem apply for all the models, or only the non-open-weights models (like GPT-4V)?
- 3(d): The authors marked 'yes' for this question, but I could not find the total amount of compute and the type of resources used in the supplemental material.
- 5(c): The authors marked 'yes', but I could not find the estimated hourly wage in the supplemental material.

**Relation To Prior Work:**

Yes.

**Summary And Contributions:**

The paper introduces a new dataset and benchmark for evaluating large multimodal agents. The benchmark is based on multi-modal murder mystery games, which involve long context windows, multi-modal inputs, and compositional reasoning, making them challenging even for SOTA models such as GPT-4V.

---

> ### Author Rebuttal · Authors · 2024-08-17
>
> Thank you for your suggestions!
> > **Q1: Citation order.**
>
> **A1**: We’ve adjusted the citation order in the bibliography as they appear in the paper.
>
> > **Q2: Lines 222: No need to say 'meticulously designed' again**
>
> **A2**: We’ve removed it.
>
> >**Q3: Line 250: 'perception questions' -> 'perceptual ability assessment questions'**
>
> **A3**: It’s indeed a more accurate phrase. We’ve changed it
>
> >**Q4: Line 282: A lot of numbers in Tables 2 and 3**
>
> **A4:** Table 2 presents the win rate statistics for agents in online battles. Each cell (Row, Col) indicates the win rate of the Row agent against the Col agent. We also evaluated the specific abilities at each stage of the game, as presented in Table 3. We uniformly report the performance of each agent in these stages when competing against a naive agent.
>
> >**Q5: Correlation between the CoE scores and the Arena Win Rates**
>
> **A5:** We refer the reviewer to General Response [Q2], where we show the relation between Table 2 and Table 3 in the attached PDF.
>
> >**Q6: The meaning of Xs and online racing**
>
> **A6:**
> - **Xs**: It indicates that these research efforts did not focus on complex reasoning, which involves cross-modal vision-text joint comprehension and multi-step reasoning.
> - **Online Racing**: It refers to a mode where agents compete against each other in real-time races. The context for agents to answer questions or make a decision is not fixed. All the agents have to adapt and respond to the actions of other players (i.e., agents) on the fly. Such an online platform is critical for assessing an agent's ability to perform in a competitive, real-time setting, akin to real-world scenarios or tasks.
>
> > **Q7: Prior benchmarks.**
>
> **A7**: Our benchmark offers an effective platform for evaluating multiple critical abilities in an integrated manner. In contrast, existing benchmarks, typically in controlled settings [2] or focused on isolated capabilities [3][4], often neglect social interaction skills [5][6], limiting their ability to assess performance in the complex dynamics of real-world scenarios.
>
> > **Q8: Experts and  Platforms (Line 114)**
>
> **A8:** "Seasoned murder mystery game experts" refers to players with extensive experience. "Industry-recognized creative teams and platforms" refers to studios and platforms with significant influence within the murder mystery game industry.
>
> > **Q9: Line 214:'Stronger'**
>
> **A9:** The term "stronger" refers to the agents in Table 3 that have relatively high scores in perception, cognition, and social interaction.
>
> >**Q10: Line 240: 'RP index'**
>
> **A10:** The calculation of the 'RP index' draws inspiration from SOTOPIA [1]. This index primarily assesses the naturalistic human-like behavior of agents. We identified several behaviors that might differentiate agents from human players, such as forgetting their role and disclosing themselves as AI, and curated the evaluation criteria.
>
> >**Q11: Line 250: 'MMR' and 'CMD'**
>
> **A11** MMR stands for Multi-step Multi-modal Reasoning and CMD stands for Crime Method and Motive Determination.
>
> >**Q12: License for the dataset.**
>
> **A12:** The curated dataset is made available for academic research purposes.
>
> >**Q13: Naive Agent.**
>
> **A13:** The Naive Agent refers to an agent lacking independent thinking and reasoning abilities. The actions/answers of the Naive Agent are predefined using human-designed template language. During discussions, this agent only retrieves information from its available script content.
>
> > **Q14 Checklist-related questions.**
> - **1(c)-Negative societal impacts:** We believe our benchmark has minimal societal impact. However, as agents integrate into daily life, the accuracy of our evaluations could shape public perception of their capabilities, possibly leading to unintended consequences.
> - **3(c)-Cost problem:**  We all use cloud computing for model inference, which incurs relatively high costs:
> Gemini 1.5 Pro: `$7/MTok(input)` , `$25/MTok(output)`; Claude: `$15/MTok(input)` , `$75/MTok(output)`.
> - **3(d)-Resources:** The primary resources we utilize in our work are the cloud services provided by the agent's company. Running all 50 scripts through a single model at once requires approximately 20 million tokens.
> - **5(c)-Estimated hourly wages:** We estimate the average hourly wage for the annotators to be approximately $8 per hour.
>
> > **Q14 Error bars.**
>
> **A14:** Due to cost considerations, the initial results were based on a single run. We have now supplemented the data with error bars for some models. We conducted three rounds of experiments.
>
> |            | Naive Agent       | Qwen           | Gemini         |
> |------------|-------------------|----------------|----------------|
> | Qwen       | 10.6±0.8          |      -     | 18.5±1.2       |
> | Gemini     | 21.5±0.7          |16.7±2.3       |     -    |
>
> ---
>
> |            | LSU               | TIU            | MIU            | MMR            | CMD            | RP             | SPC            | ITD            |
> |------------|-------------------|----------------|----------------|----------------|----------------|----------------|----------------|----------------|
> | Qwen       | 38.80±0.23        | 47.12±2.06     | 39.11±0.19     | 16.50±0.59     | 15.32±0.34     | 6.37±0.41      | 1.62±0.7       | 16.71±1.10     |
> | Gemini     | 92.0±0.23         | 66.30±0.90     | 57.03±1.23     | 55.69±0.72     | 19.38±1.16     | 7.45±0.11      | 10.64±0.08    | 26.86±0.88     |
>
> **Reference.**
>
> [1] Sotopia: Interactive evaluation for social intelligence in language agents.ICLR,2024.
>
> [2] Look before you leap: Unveiling the power of gpt-4v in robotic vision-language planning.
>
> [3] Travelplanner: A benchmark for real-world planning with language agents. ICML,2024
>
> [4] Avalonbench: Evaluating llms playing the game of avalon. NeurIPS 2023 Workshop.
>
> [5] Gaia: a benchmark for general ai assistants. ICLR,2024
>
> [6] Mobile-agent: Autonomous multi-modal mobile device agent with visual perception.ICLR,2024

---

> > ### Comment · Reviewer_7EJe · 2024-08-29
> > **Response to rebuttal**
> >
> > Thank you for your response.
> >
> > Some follow-up questions below:
> >
> > > We estimate the average hourly wage for the annotators to be approximately $8 per hour.
> >
> > How does $8/hour compare to the minimum wage for the region where the annotators live?
> >
> > > We have now supplemented the data with error bars for some models.
> >
> > Do these error bars represent the standard deviation, standard error of the mean, or something else?

---

> > > ### Author Response · Authors · 2024-08-30
> > >
> > > Dear Reviewer 7EJe,
> > >
> > > Thank you for your insightful question.
> > >
> > > 1. According to the latest data, the wages in our region are shown as follows:
> > >
> > > * minimum wage: $3 per hour
> > >
> > > * average wage: $5 per hour
> > >
> > > * stipend for Ph.D. students: ~$5 per hour
> > >
> > > 2. The error bars represent the mean ± standard deviation, calculated from data obtained across  three experimental runs.
> > >
> > > Thank you for your consideration. We are happy to address any further questions you may have.
> > >
> > > Best wishes,
> > >
> > > Authors

---

> > > > ### Comment · Reviewer_7EJe · 2024-08-30
> > > > **Response to further clarifications**
> > > >
> > > > Thank you for your answers. I suggest including this information in the final version of the manuscript, even if it is in the appendices. I will raise my score by one point.

---

### Author Rebuttal · Authors · 2024-08-17

Dear Reviewers,

We would like to express our gratitude to all reviewers for their thoughtful and constructive feedback. We were encouraged to see the reviewers rated it a novel (Reviewers 7EJe, NGnP, PMa7) and interesting (Reviewers NGnP and 1p2a) study. The reviewers also appreciated how our benchmark tests a large number of agents (Reviewer 7EJe) in a real-world setting (Reviewers NGnP, 1p2a). We have been diligently working on addressing the concerns and suggestions of the reviewers.

**Q1. Exploration of human-agent interaction (Reviewers PMa7)**

**A1:** We would like to thank the reviewers for this great idea. Our goal of this study is to propose a real-world benchmark to automatically benchmark different multi-modal agents. We believe that the idea of human-agent interaction would significantly broaden the impact of this paper if people could “play” with our benchmark. Therefore, we started to develop an interface that enables human participants to directly join and engage with agents in a murder mystery game scenario. The prototype is shown in the attached PDF. It can serve as a tool to (1) make researchers better understand our benchmark and (2) allow people to rate the performance of different agents like LMSYS Chatbot Arena. Besides, The human-agent interaction data allows for in-depth analysis of human/agent behavior patterns, decision-making processes, and the effectiveness of human-agent collaboration, providing valuable insights for the future development of intelligent systems.

**Q2: More qualitative analysis (e.g., visualization) (Reviewers 7EJe,NGnP,1p2a)**

**A2:** In our work, we introduce a comprehensive benchmark via murder mystery games designed to evaluate LMAs (named WhodunitBench). We propose two evaluation modes:

- **Arena-style Evaluation.** Two types of agents participate in the murder mystery game, each playing the roles of either non-murderer characters or the murderer. Table 2 presents the win rate statistics for agents in this mode. Each cell (Row, Col) indicates the win rate of the Row agent against the Col agent, where the Row agent plays the role of the non-murderer, and the Col agent assumes the role of the murderer.

- **Chain of Evaluation (CoE)**. To provide a more in-depth and interpretable analysis of the agents' performance, in addition to the final win rate, we also evaluated the specific abilities demonstrated by the models at each stage of the game, including perception, cognition, role-play and decision-making, as presented in Figure 1 (in our paper). We uniformly assess such capabilities of each agent when competing against a naive agent. And the evaluated results are reported in Table 3.

We have conducted a visual analysis of the relationships between the metrics in Tables 2 and 3 (as shown in the attached PDF). The results demonstrate a linear relationship between win rate and the average CoE score, indicating that as the CoE average score increases, the agent's performance improves, resulting in a higher likelihood of winning matches. Among the CoE indicators, reasoning-related metrics exhibit the strongest correlation with win rate, suggesting that reasoning capabilities are the most significant contributors to success.

---

### Decision · Program_Chairs · 2024-09-26

**Decision:**

Accept (Spotlight)

**Comment:**

This paper introduces a new benchmark for multimodal agents based on murder-mystery games. Mechanistically, such games typically involve a group of players role-playing based on a script, where they must elicit clues from each other (including the assigned ‘murderer’ who tries to hide his steps) and reason and vote to deduce the murderer. The game often requires long-horizon multi-step reasoning, multi-modal perception, goal oriented behaviours and benefits from human-style “theory of mind”. In this case the benchmark provides two settings: an “Arena” style evaluation, where models compete against each other, and a “Chain of Evaluation” multiple-choice question evaluation, wherein different skills (such as language perception, or strategic decision making) are tested. The paper presents the performance of the leading models on the benchmark including GPT4-V, Claude Opus, Gemini  and others.


**Strengths:**
* Reviewers commented on the good motivations for the task, as well as the originality of the work, and the potential challenge of the task. Many reviewers seemed excited or intrigued by the skills required to be good at this task.
* Reviewers commented on the substantial work on the CoE evaluation, that allows researchers to disentangle the various axes an agent must improve along, to be proficient at the task.

**Weaknesses:**
* Many reviewers mentioned that the benchmark would be strengthened by including comparisons to human (‘real’) performance. This is considerable limitation of the paper, and fails to ground a realistic ‘upper bound’ on performance to the casual reader. In rebuttals however, authors noted how this would broaden the impact of the work, and started to create a web interface for humans to play the game (visible in screen shots). They also added some small-sample human assessments and said they would add them to the paper.
* The authors had a very short section on limitations that multiple reviewers suggested should be augmented to give a more complete assessment of limitations.

**AC Note:**
* Following the unanimous recommendation of the review, I recommend this for acceptance. Success at the murder mystery game indeed requires a number of different axes of intelligence/agency, and this benchmark allows researchers to both analys these skills holistically (in the arena setting) and ablate performance with the CoE evaluations. I expect with the advent of ‘reasoning models’ like OpenAI’s o1, such ‘head to head’ evaluations will only prove more impactful.